# Chronic circadian disruption modulates breast cancer stemness and immune microenvironment to drive metastasis in mice

Eva Hadadi [1✉], William Taylor[1], Xiao-Mei Li[1,2], Yetki Aslan[3], Marthe Villote[4], Julie Rivière[4], Gaelle Duvallet[5], Charlotte Auriau[5], Sandrine Dulong [1,2], Isabelle Raymond-Letron [6,7], Sylvain Provot [3], Annelise Bennaceur-Griscelli[1,2,8] & Hervé Acloque [1,4✉]

Breast cancer is the most common type of cancer worldwide and one of the major causes of cancer death in women. Epidemiological studies have established a link between night-shift work and increased cancer risk, suggesting that circadian disruption may play a role in carcinogenesis. Here, we aim to shed light on the effect of chronic jetlag (JL) on mammary tumour development. To do this, we use a mouse model of spontaneous mammary tumourigenesis and subject it to chronic circadian disruption. We observe that circadian disruption significantly increases cancer-cell dissemination and lung metastasis. It also enhances the stemness and tumour-initiating potential of tumour cells and creates an immunosuppressive shift in the tumour microenvironment. Finally, our results suggest that the use of a CXCR2 inhibitor could correct the effect of JL on cancer-cell dissemination and metastasis. Altogether, our data provide a conceptual framework to better understand and manage the effects of chronic circadian disruption on breast cancer progression.

[1] Inserm, U935, Université Paris Sud, Villejuif, France. [2] Université Paris Sud, Université Paris Saclay, UFR de Médecine Kremlin Bicêtre, Le Kremlin-Bicêtre, France. [3] Inserm, U1132, Université Paris Diderot, Hôpital Lariboisière - Centre Viggo Petersen, 75010 Paris, France. [4] GABI, INRA, AgroParisTech, Université Paris-Saclay, 78352 Jouy-en-Josas, France. [5] Inserm, UMS33 Villejuif, France. [6] Département des Sciences Biologiques et Fonctionnelles, Laboratoire d'HistoPathologie Expérimentale et Comparée (LabHPEC), ENVT, Université de Toulouse, Toulouse, France. [7] STROMALab, CNRS ERL5311, EFS, ENVT, Inserm U1031, Université de Toulouse, Toulouse, France. [8] Service d'hématologie, APHP, GHU Paris Sud, Paris, France. ✉email: eva.hadadi@inserm.fr; herve.acloque@inra.fr

Globally, breast cancer (BC) is the most frequent cancer in women. The cumulative risk that a woman will develop BC is around 5% worldwide, with a 1.4% risk of death. In 2018, there were more than 2 million newly diagnosed cases, representing almost 25% of all cancer cases in women. BC is the leading cause of death for women in most countries[1]. Genetic causes account for <10% of BC; instead, the majority of BC development has been linked with non-hereditary causes. These include nutrition-related factors, alcohol consumption, exogenous hormone intake, reproductive history, and menstruation parameters[1]. Environmental factors such as air pollution or altered light/dark cycles, such as those experienced by night-shift workers, can also affect BC incidence[2–4]. Indeed, in 2007 the International Agency for Research on Cancer (IARC) classified circadian rhythm disruption (CRD) as probably carcinogenic based on eight epidemiological studies and data from animal models[4]. Since then, additional and better-documented epidemiological studies, genome-wide association studies (GWAS), and cellular and animal studies have substantiated the link between BC development and circadian disruption[5]. A recent population-based case-control study confirmed that factors including night-work duration, length of shifts, and time since last night shift affect the odd ratios for BC, mostly in premenopausal women[6]. In this, hormonal receptor status also plays an important role: BC risk associated with night work is only higher for ER+HER2+ cancer. These epidemiological studies are supported by GWAS analyses that have revealed a significant statistical association between genetic variation located in circadian genes (*ARNTL, CLOCK, CRY1, CRY2, RORA, RORB, RORC, PER1*) and the risk of breast cancer, as well as between circadian clock-gene expression and metastasis-free survival[7,8]. Experimental studies on mammary epithelial cells have also provided evidence of an important role for core circadian clock genes in mammary gland formation and function. Specifically, female *Per2*$^{-/-}$ mutant mice fail to form normal terminal mammary ducts, and instead have an excess of basal progenitors[9]. Female *Arntl*$^{-/-}$ mutants have fewer ductal branches, shorter ductal length and more terminal end buds, while female *Clock*$^{-/-}$ mutants present defects in daytime maternal behaviour and milk production[10,11]. Studies of core circadian genes in human mammary epithelial cells have confirmed these in vivo observations and have identified a strong effect of these genes on the stemness of mammary epithelial cells, either by decreasing or increasing stemness[9,12,13]. However, it is not always clear whether the observed phenotypes result from the non-circadian function of these transcription factors or are instead an indirect consequence of global dysregulation of the circadian clock in cells and tissues.

Experimental results on the effects of CRD on breast cancer onset have already been obtained using an inducible *p53* mutant mouse model that was predisposed to developing primary mammary tumours. These mice typically developed mammary tumours in 50 weeks, but onset was 8 weeks earlier when the mice experienced CRD. This study supported the key role that CRD can have in driving breast cancer development[14].

Here, we wanted to progress beyond the onset of tumourigenesis and explore the effects of CRD on tumour progression, cancer-cell dissemination, and immune phenotype. To do this, we use the MMTV:PyMT model of spontaneous murine mammary carcinogenesis[15] and test the effects of chronic CRD applied for 10 weeks at the beginning of puberty-initiated tumourigenesis. We observe that circadian disruption significantly increases cancer-cell dissemination and metastasis by acting on the stemness and tumour-initiating potential of tumour cells and by creating an immunosuppressive shift in the tumour microenvironment.

## Results

### Chronic CRD moderately affects primary tumour development.

The original MMTV:PyMT mouse model with FVB background (FVB PyMT) is known to experience rapid and strong metastasis[15]; instead, the PyMT mouse model with the C57Bl/6J (B6) background experiences delayed tumourigenesis, with a more gradual but variable tumour growth rate and reduced lung metastasis compared with FVB PyMT mice[16]. As the aggressive FVB PyMT model was incompatible with our goal of modelling long-term chronic CRD, and because we were also interested in investigating the earlier/linear phase of tumour growth, we decided to use a mixed B6*FVB PyMT background. In line with previous observations, we observed a delayed onset of tumour development and slower cancer progression in these mice compared with typical FVB mice, with a low prevalence (ca. 30%) of lung metastasis at the age of 16 weeks. To maximise the possibility of observing differences between our experimental groups, mice were analysed in the early/mid phase of tumour development, from 6 to 16 weeks (Fig. 1a). At 6 weeks old, mice carrying the MMTV:PyMT and MMTV:LUC transgenes were divided into two lots. One was maintained for 10 weeks in normal conditions of alternating light and dark periods (LD, 12-h light and 12-h dark), while the other group was exposed for 10 weeks to chronic CRD through continuous jet lag, simulated by a reduction of 8 h in the dark period every other day (JL) (Fig. 1a). This jet-lag protocol was previously shown to totally disrupt the 24-h periodicity of the rhythmic pattern of rest (12-h light period) and activity (12-h dark period) and is frequently used to mimic the effects of shift work or frequent eastbound transmeridian flights[17]. By assessing the locomotor activity and body temperature of these mice, we confirmed that this jet-lag protocol also disrupted circadian activity in our transgenic model (Fig. 1b, Supplementary Fig. 1 and Supplementary Table 1). From 10 weeks of age to the end of the experiment at 16 weeks, the growth of primary tumours was monitored every 2 weeks by imaging the luciferase activity derived from the MMTV:LUC transgene. At 16 weeks of age, mice were weighed and sacrificed, and their tissues were processed for further analysis. The median weight of JL mice was slightly but not significantly higher than that of LD mice, as expected from previous studies (Fig. 1c)[18,19]. Peripheral blood cell counts were similar between the two experimental conditions (Fig. 1d). In addition, there were only a few differences in the blood biochemistry parameters of the two groups of mice: glucose level was significantly reduced in JL mice, while total levels of cholesterol, triglycerides and free fatty acids were significantly increased in the plasma of these mice (Fig. 1e and supplementary Fig. 2). In vivo imaging revealed no significant difference in the onset of tumourigenesis (Fig. 1f). However, in JL mice, in vivo bioluminescence measurement of tumour growth showed a slight increase with age, and tumour burden was significantly higher (Fig. 1f, g). We assessed the grade of primary tumours on paraffin-embedded sections stained with HES (hematoxylin, eosin, and saffron). Multiple tumour grades, ranging from hyperplasia to late carcinoma[20], were observed in primary tumours from both LD and JL mice, but, overall, the lesions in JL mice were more malignant (Supplementary Table 2 and Supplementary Fig. 3).

### CRD promotes cancer-cell dissemination and metastasis.

Using these mice, we then explored if chronic CRD affected the dissemination of cancer cells. We quantified the presence of disseminated cancer cells (DCCs) using flow cytometry and real-time PCR analyses of the expression of the PyMT transgene. With both types of analyses, we observed a significant elevation of transgene expression in the bone marrow (BM) of JL mice, with

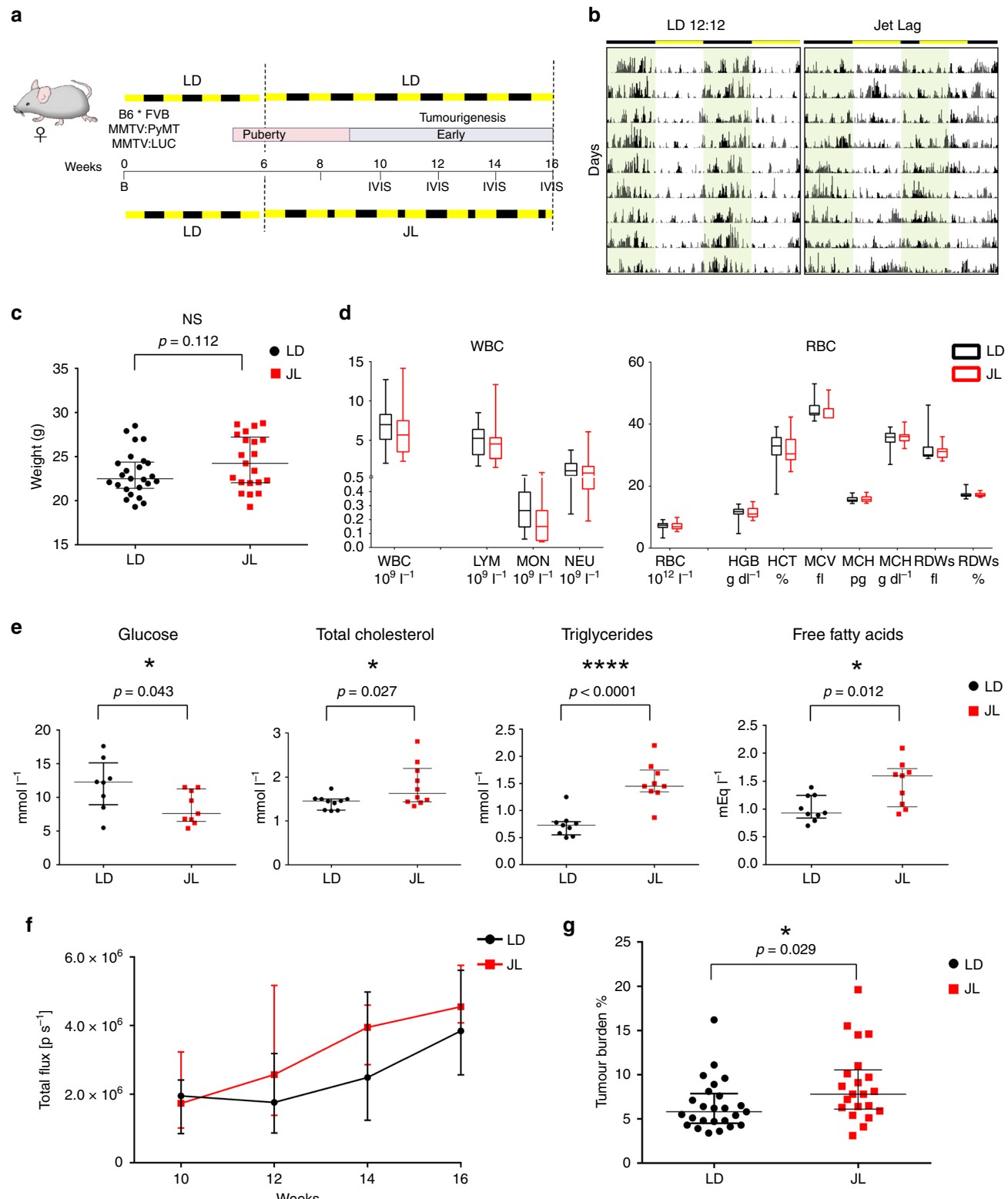

an almost two-fold increase in DCCs in the Lin⁻ mononucleated cells of BM (Fig. 2a). Flow analysis also confirmed an increase in circulating cancer cells (CTCs)in the bloodstream of JL mice (Fig. 2b). Furthermore, we observed DCC cells with H&E staining of bone sections, and μCT analysis also revealed bone lesions, highlighting the dissemination of cancer cells to bone (Supplementary Fig. 4a). Consistent with these observations, we observed

a significant increase in the prevalence of metastasis in JL compared with LD mice (Fig. 2c). The proportion of mice with lung metastasis increased from 28% in LD to 52% in JL (Fig. 2c) and the number of metastatic foci was also significantly higher in JL mice (Fig. 2d; Supplementary Fig. 4b). Altogether, our results reveal a significant impact of chronic CRD on cancer-cell dissemination and metastasis.

**Fig. 1 Chronic circadian disruption slightly enhances tumour burden. a** Experimental timeline for evaluation of the effect of chronic jet lag on spontaneous mammary tumourigenesis in B6*FVB PyMT mice. B: birth; IVIS: tumour growth monitoring using bioluminescence. Dashed lines highlight the start and end points of the experiment. **b** Representative actograms of LD and JL mice. LD: 12-h light and 12-h dark; JL: jet lag, represented by a shortening of the dark period by 8 h every second day. **c** Weight at sacrifice of mice in LD ($n = 25$) and JL ($n = 21$) conditions. Data are presented as a scatter dot plot with lines indicating the median with interquartile range (error bars). $P$-value calculated from an unpaired two-sided $t$-test. **d** Blood cell counts: total numbers of white blood cells (WBC) and red blood cells (RBC) in LD ($n = 16$) and JL ($n = 17$) mice. LYM lymphocytes, MON monocytes, NEU neutrophils, HGB haemoglobin, HCT haematocrit, MCV mean corpuscular volume, MCH mean corpuscular haemoglobin, RDW red cell distribution width. Data presented as box-and-whisker plots. Variability is shown using medians (line in the box), 25th and 75th percentiles (box), and min to max (whiskers). **e** Glucose ($n = 8$ LD and $n = 9$ JL), total cholesterol ($n = 10$ LD and $n = 10$ JL), triglyceride ($n = 9$ LD and $n = 9$ JL, $p = 5.04882e{-}5$), and free fatty acid ($n = 9$ LD and $n = 9$ JL) profiles of LD and JL mice. Data are presented as scatter dot plots with lines indicating the median with interquartile range (error bars). $P$-value calculated from an unpaired two-sided $t$-test. **f** Timelines of tumour growth in total flux [p s$^{-1}$] measured by in vivo bioluminescence imaging in LD ($n = 6$) or JL ($n = 5$) groups. Data are presented as a dot plot with dots indicating the median with interquartile range (error bars). **g** Tumour burden (tumour to body weight ratio) as % in LD ($n = 25$) or JL ($n = 21$) conditions. Data are presented as a scatter dot plot with lines indicating the median with interquartile range (error bars). $P$-value calculated from an unpaired two-sided $t$-test. Indicated ($n$) represent number of independent experiments as biological replicates.

**CRD modulates expression of phototransduction genes**. In order to identify the potential molecular players and pathways that drove the increase in dissemination in JL mice, we performed an mRNA-seq study on bulk of dissociated primary tumour cells and on bulk of BM mononuclear cells from five JL and five LD mice (Supplementary Fig. 5a). Hierarchical clustering based on global gene expression profiles clearly separated BM and primary tumour samples, but did not distinguish between LD and JL conditions or the absence/presence of metastasis in lungs (Fig. 3a). We confirmed this observation through principal components analysis on BM and primary cancer cells (Supplementary Fig. 5b), in which the two first principal components axes did not separate LD and JL samples. We then performed differential gene expression analysis. As expected from the hierarchical clustering analysis, only a few genes were significantly differentially expressed between JL and LD conditions (Supplementary Data 1 and Supplementary Data 2). Intriguingly, in mononuclear BM cells, the genes with the largest differences in expression (*Rhodopsin, Gnat1, Rbp3* and *Prph2*) were associated with Gene Ontology (GO) terms linked with photoperception and phototransduction (Fig. 3b). Compared with LD controls, these genes were strongly downregulated in mononuclear BM cells from JL mice: *Rhodopsin* by 23-fold, *Gnat1* by 11-fold, *Rbp3* by 7-fold and *Prph2* by 6-fold (Supplementary Data 1). Similarly, in primary tumours these genes showed significant down-regulation (3–4-fold), together with other genes associated with light perception and phototransduction, such as *Cngb1, Nrl* (6-fold) or *Bhlhe40* (Supplementary Data 2). Hierarchical clustering using the genes belonging to the GO term phototransduction clearly separates BM samples from JL and LD mice while a subset of these genes also clearly separates primary tumour samples from JL and LD (Fig. 3c and Supplementary Fig. 5c). These data suggest a strong effect of CRD on the expression of genes whose function is associated with light perception and light signal transduction, even in tissues that are not directly exposed to light (due to poor tissue penetration) or not known to respond to light stimulation.

**CRD promotes stemness of primary tumour cells**. In order to understand how chronic CRD can affect the dissemination and metastatic potential of mammary cancer cells, we further characterised the cellular composition of primary tumours. We first tried to assess whether the proportion of mammary cancer stem cells was different between LD and JL conditions. To do this, we quantified the expression of known markers of mammary cancer-cell stemness from primary tumours. Using flow cytometry, we observed a statistically significant increase in the percentage of cancer cells that were positive for CD24, CD49f,

and CD326 in JL mice (Supplementary Fig. 6a). Similarly, we observed an increase in the expression of *Itgb1* (coding for CD29) and *Itga6* (coding for CD49f) (Supplementary Fig. 6b and Supplementary Data 3), together with the upregulation of genes associated with EMT (Supplementary Fig. 6c and Supplementary Data 3), a key biological process also known to modulate the stemness of breast cancer cells[21,22]. Moreover, the expression of *Inhibin-βA* (*Inhba*) was four times higher in the primary tumours of JL mice. *Inhba* encodes a protein subunit that is necessary to activate TGFβ signalling, a key pathway associated with EMT[23]. We then evaluated the proportions of cancer stem cells using the mouse mammary stem-cell (MaSC) signature CD24$^{med}$CD29$^{hi}$CD49f$^{hi}$ (Fig. 4a), which has also been described for mouse mammary cancer stem cells[24–27]. We observed a significant enrichment in CD24$^{med}$CD29$^{hi}$CD49f$^{hi}$ cancer cells in the primary tumours of JL mice (Fig. 4b). Next, we tested whether this enrichment was associated with an actual increase in the stemness potential of primary tumours. To test this hypothesis, we purified cancer cells from primary tumours of LD and JL mice (see Methods) and performed mammosphere-formation assays. Mammosphere-formation efficiency (MFE) was significantly higher in cancer cells from primary tumours of JL mice (Fig. 4c). Previous studies showed that *Per2* plays an important role in tumour suppression as its knockdown increases the stemness of mammary epithelial and cancer cells[13,28]. We also observed a decrease of *Per2* and *Cry2* expression in primary tumours of JL mice (Fig. 4d and Supplementary Data 3) while expression levels of other core clock genes remained similar between treatments (Fig. 4d and Supplementary Data 3). Furthermore, we observed that circadian oscillations of clock genes modulated the stemness of human mammary epithelial cells. Specifically, we synchronised MCF12A cells with respect to their circadian rhythm and sampled cells at times that corresponded to peaks in *PER2* or *BMAL1* expression (Supplementary Fig. 6d, e)[29]. We then assessed the stemness of these cells at the different time points. We observed a diurnal oscillation in mammosphere-formation efficiency, with a negative correlation with the expression peaks of *PER* genes (Fig. 4e). Finally, to quantify the tumour-initiation potential of cells from the primary tumours, we performed orthotopic injection of isolated tumour cells into the mammary fat pads of recipient mice (see Methods). We found that grafted cancer cells that were purified from JL donor mice demonstrated increased tumour-initiating potential in immunocompetent wild-type C57BL/6J mice compared with cells obtained from LD donors (Fig. 4f). We further confirmed that the tumours used for transplants and mammospheres formation were all of similar subtypes, ruling out the possibility that

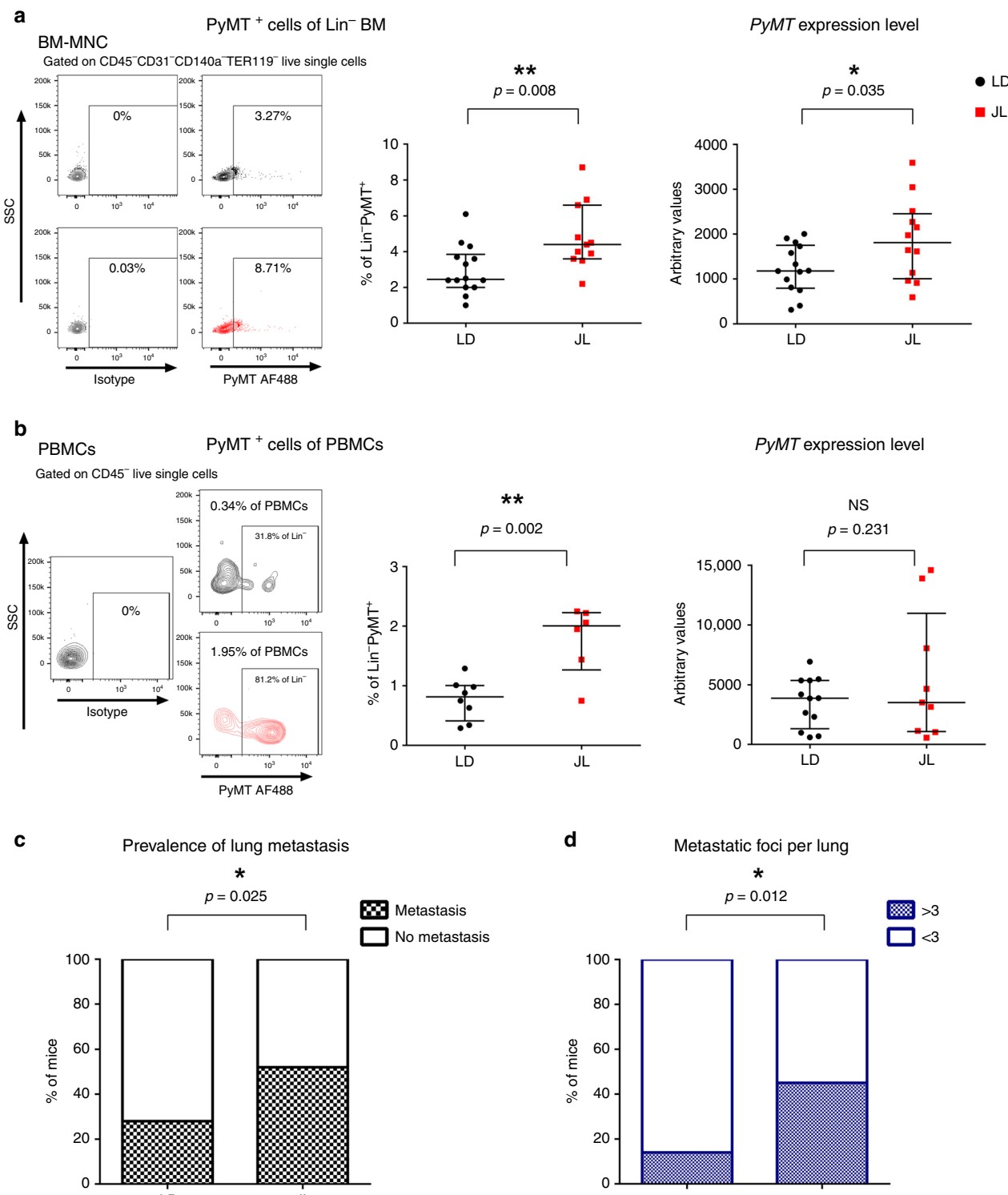

**Fig. 2 Chronic circadian disruption increases cancer-cell dissemination and metastasis. a** Disseminated tumour cells detected in LD and JL mice at 16 weeks of age by flow cytometry (left, LD $n = 14$ and JL $n = 11$) and real-time PCR (right, LD $n = 14$ and JL $n = 12$) of bone marrow mononucleated cells (BM-MNC). Representative flow cytometry plots showing intracellular PyMT staining of CD45⁻CD31⁻CD140a⁻Ter119⁻ live BM-MNC cells. Data are presented as scatter dot plots with lines indicating the median with interquartile range (error bars). *P*-values are calculated from an unpaired two-sided *t*-test. **b** Circulating tumour cells in peripheral blood mononuclear cells (PBMCs) detected by flow cytometry (left; LD: $n = 8$ and JL: $n = 6$) and real-time qPCR (right; LD: $n = 12$ and JL: $n = 9$) in mice at 16 weeks of age. Representative flow cytometry plots showing intracellular PyMT staining of CD45⁻ live PBMCs (SSC: side scatter). Data are presented as scatter dot plots with lines indicating the median with interquartile range (error bars). *P*-values are calculated from an unpaired two-sided *t*-test. **c** Prevalence of lung metastasis in LD ($n = 25$) and JL ($n = 21$) cohorts at 16 weeks of age. Data represent the percentage of mice with metastasis in both conditions, *P*-value obtained from a binomial two-sided test. **d** Number of metastatic foci per lung in LD ($n = 25$) and JL ($n = 21$) mice at 16 weeks of age. Data represent the percentage of mice with >3 or <3 metastatic foci in both conditions. *P*-value obtained from a binomial two-sided test. Indicated (*n*) represents the number of independent experiments as biological replicates.

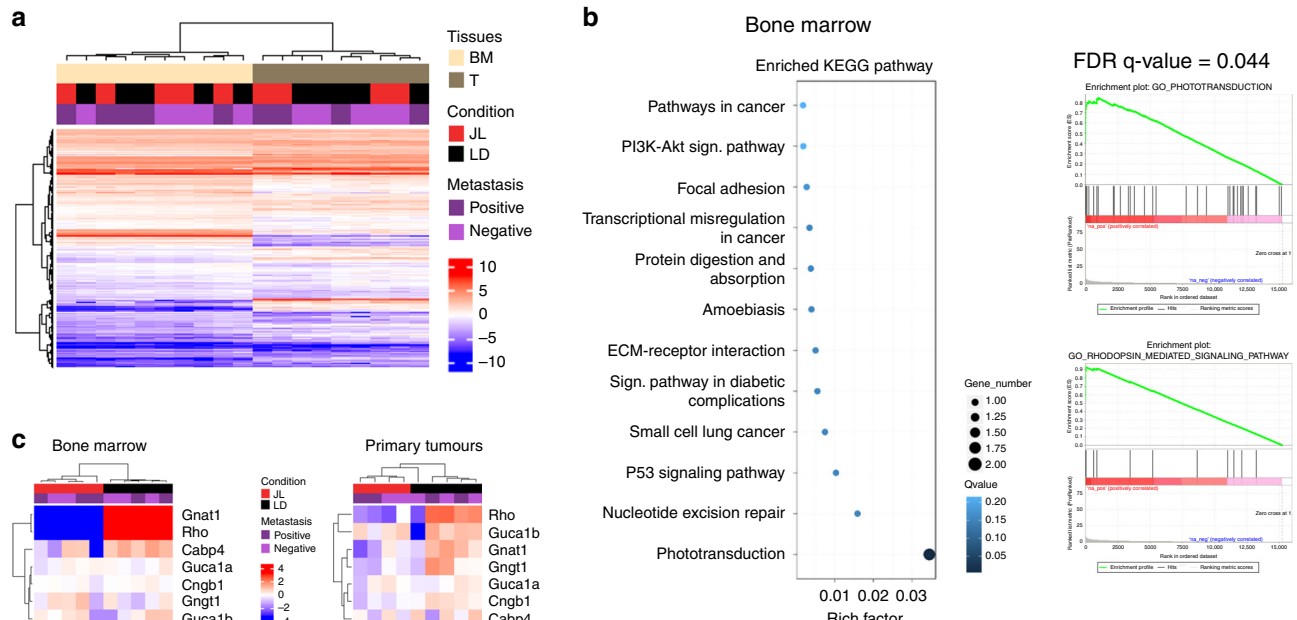

**Fig. 3 Chronic circadian disruption does not profoundly alter gene expression profiles in primary tumours or bone marrow mononucleated cells, with the exception of genes linked with phototransduction and light perception. a** RNA-seq sample heatmap and hierarchical clustering based on the expression (FPKM) of 12,556 genes across the two tissues. Gene expression matrix was centered, reduced, and log2 transformed. Samples appear as columns and genes as rows; samples are labelled by their tissue of origin (BM: bone marrow ($n = 10$, salmon) and T: primary tumour ($n = 9$, grey)), experimental conditions (JL ($n = 9$) in red and LD ($n = 10$) in black) and the presence ($n = 9$)/absence ($n = 10$) of metastasis (dark and light violet). Hierarchical clustering was performed using Euclidean distance and the Ward.D2 criterion for agglomeration. The colour scale represents expression levels, red for high expression and blue for low expression. **b** Enriched KEGG Pathway and Gene Set enrichment analysis for bone marrow. GSEA plots are shown for the Gene Ontology (GO) terms: Phototransduction and Rhodopsin_mediated_signaling_pathway. **c** Heatmap based on the expression (FPKM) of differentially expressed genes linked with phototransduction and photoperception in bone marrow ($n = 10$) and primary tumours ($n = 9$). Gene expression matrix was centered, reduced, and log2 transformed. Samples appear as columns and genes as rows; samples are labelled by the experimental conditions (JL in red and LD in black) and the presence/absence of metastasis. Hierarchical clustering was performed using Euclidean distance and the Ward.D2 criterion for agglomeration. The colour scale represents expression levels, red for high expression and blue for low expression. Indicated ($n$) represents the number of independent experiments as biological replicates.

this may be the cause of the observed differences in MFE and tumour-initiating potential (Supplementary Table 2).

Altogether, these data support the findings of previous studies, but also provide further evidence for the importance of a functional circadian clock in regulating the stemness of mammary epithelial cells and deepen our understanding of the biological consequences of CRD.

**CRD promotes an immunosuppressive tumour microenvironment.** The immune system plays a critical role in tumour progression and metastatic dissemination. To investigate whether the increased prevalence of metastasis in JL mice resulted from changes in the immune microenvironment of tumours, we characterised tumour-infiltrating cells (TICs) using flow cytometry (see the representative gating strategy[30] in the Supplementary Fig. 7). We found reduced numbers of CD45$^+$ immune cells in tumours from JL mice (Fig. 5a), but no significant alteration in the proportional distribution of different immune cell types (Fig. 5b; Supplementary Fig. 8). However, fine-scale characterisation of subpopulations of macrophages and T cells revealed significant differences between the LD and JL tumour immune microenvironment. Specifically, the CD64$^+$CD24$^-$ macrophages, determined by using the gating strategy described by Yu et al.[30], were further dissected based on their MHC II expression level. Similarly to previous report[31], we identified CD11b$^+$MHC II$^{hi}$ anti-tumour and CD11b$^+$MHC II$^{low}$ pro-tumour tumour-associated macrophages (TAMs) (Supplementary Fig. 7a). In JL tumours, the

percentage of tumour-suppressive MHC II$^{hi}$ TAMs was not significantly altered compared with LD tumours (Fig. 5c). Instead, JL tumours had a significantly higher proportion of tumour-supporting MHC II$^{low}$ TAMs (Fig. 5d). Characterization of the major T-cell populations revealed a decrease in the number of infiltrating CD8$^+$ T cells and an increase in the CD4/CD8 ratio in JL tumours (Fig. 5e). Importantly, we observed a significant enrichment of the immunosuppressive CD4$^+$FoxP3$^+$ Treg population in the primary tumours of JL mice and consequent elevation of Treg/CD8 ratio (Fig. 5f), which together with CD4/CD8 ratio is a prognostic indicator of therapy responsiveness and survival in breast cancer patients[32–34]. A significant elevation in the CD4/CD8 ratio was also detected in the peripheral blood of JL mice (Fig. 5g, Supplementary Fig. 8b). This additional use of CD4 or Treg to CD8 ratio also provides cleaner representation of TIL/lymphocyte values, which in general show high inter-individual variability (Fig. 5e, f, Supplementary Fig. 8b). Our results suggest that chronic CRD weakens the anti-tumour immune response and creates an immunosuppressive pro-tumour microenvironment. The latter effect, together with the effects of CRD on cancer-cell stemness, may help to facilitate the dissemination of mammary cancer cells and the formation of lung metastasis.

**Chronic CRD alters the cytokine–chemokine network.** The importance of circadian regulation in leukocyte homoeostasis and migration is well-documented[35] to better understand how chronic CRD could modify the immune microenvironment of

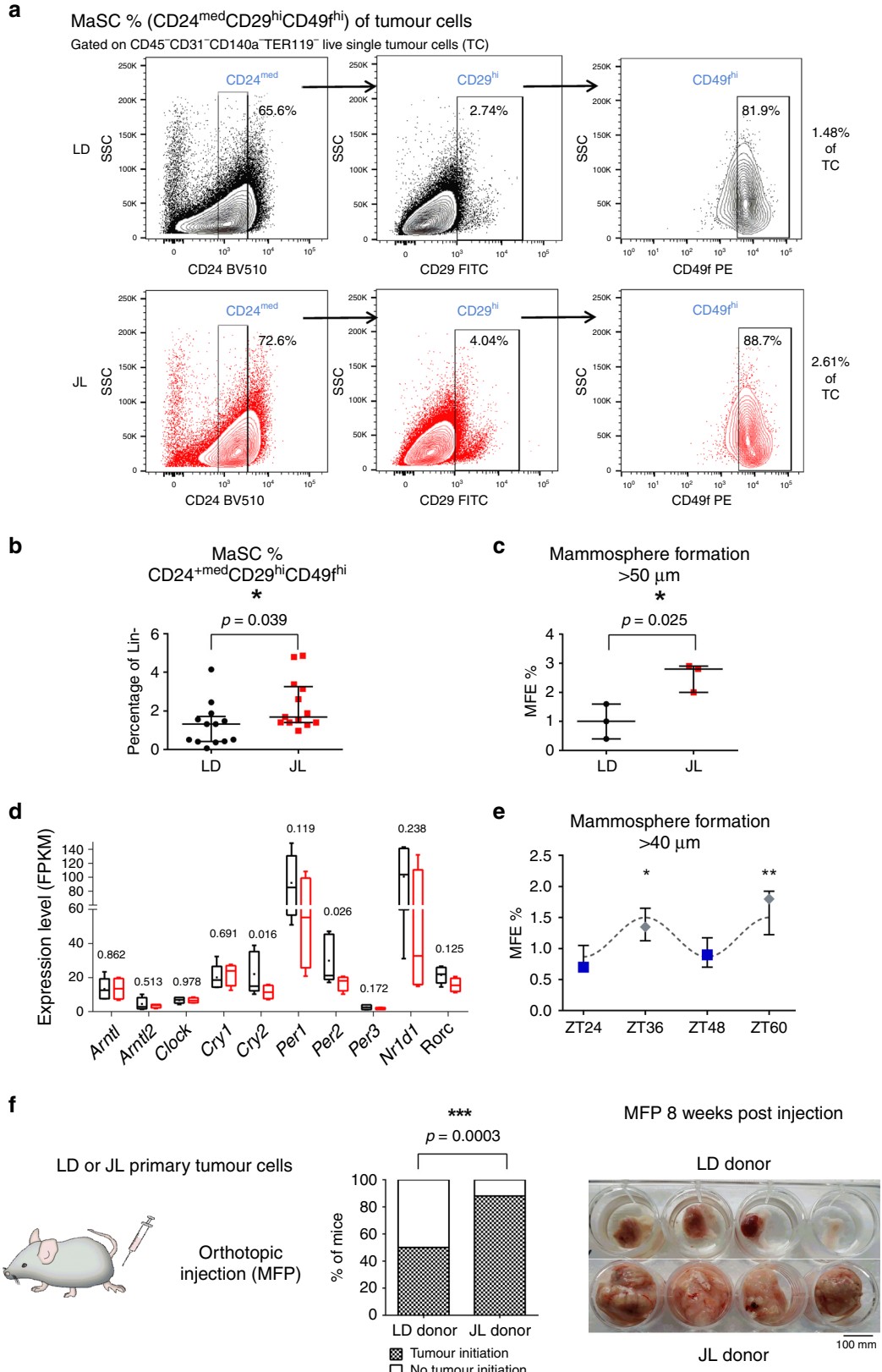

tumours, we decided to investigate the cytokine–chemokine network in JL mice. Using a magnetic Luminex assay, we first quantified levels of 17 circulating cytokines in plasma from JL and LD mice. No significant differences were observed, with the exception of IL-4 whose circulating levels were slightly lower in

JL mice (Supplementary Table 3). Since plasma cytokine/chemokine levels are not necessarily representing the tumour microenvironment, we used the data of our transcriptomic study and real-time PCR to assess the expression levels of cytokines/chemokines and their receptors in primary tumours.

**Fig. 4 Chronic circadian disruption increases cancer-cell stemness in primary tumours. a** Representative gating strategies for CD24$^{med}$CD29$^{hi}$CD49f$^{hi}$ mammmary stem cells (MaSC) with contour plots shown for LD (black) and JL (red) tumours (SSC: side scatter). **b** Frequency of CD24$^{med}$CD29$^{hi}$CD49f$^{hi}$ mammary stem cells (MaSC) in LD and JL tumours ($n = 13$). Data are shown as a scatter dot plot with lines indicating the median with interquartile range (error bars). $P$-value obtained from an unpaired two-sided $t$-test. **c** Mammosphere-formation efficiency (MFE%) of LD and JL tumour cells ($n = 3$). Data are shown as a scatter dot plot with lines indicating the median with interquartile range (error bars). $P$-value obtained from an unpaired two-sided $t$-test. **d** Circadian clock genes mRNA expressions (FPKM: Fragments Per Kilobase Million) in cancer cells from LD ($n = 5$) and JL ($n = 4$) primary tumours. Data presented as box-and-whisker plots. Variability is shown using medians (line in the box), 25th and 75th percentiles (box), and min to max (whiskers). $P$-values were calculated using DESeq2 and the Wald test based on the negative binomial distribution. Respective log2FoldChange (log2FC) are listed in Supplementary Data 3. **e** Mammosphere-formation efficiency (MFE%) of MCF12A normal human mammary cells in different circadian phases ($n = 3$). Blue squares and grey diamonds represent peaks of *BMAL1*$^{HIGH}$/*PER2*$^{LOW}$ and *BMAL1*$^{LOW}$/*PER2*$^{HIGH}$ expression, respectively. Data are presented as a dot plot with dots indicating the median with interquartile range (error bars). $P$-value was calculated from a one-way ANOVA/Tukey's multiple comparisons test. **f** Tumour-initiation study based on orthotopic injection of primary tumour cells from LD ($n = 6$) and JL ($n = 6$) mice in the mammary fat pad (MFP) of host mice ($n = 4$ per each donor). Tumour-initiation potential was calculated as the percentage of host mice that formed tumours. We set a threshold based on MFP weight at 0.25 g for positivity. $P$-value was obtained from a binomial two-sided test. The picture illustrates the observed differences between MFP between JL and LD donors. Indicated ($n$) represent number of independent experiments as biological replicates.

We observed that the most downregulated ($< -0.8$ Log2FC) cytokines/chemokines in JL primary tumours, including *Ifng, Cxcl13, Tnfs18* and *Cxcl11*, are known to favour an anti-tumour immune response (Fig. 6a; Supplementary Data 3). While the other hand, the most upregulated ($> 0.8$ Log2FC) ones, including *Cxcl3, Cxcl5, Il10* and *Il1b*, are linked to immuno-suppression or tumour progression (Fig. 6a, Supplementary Data 3; Supplementary Fig. 9b). In BM mononucleated cells, transcriptomic analysis revealed a high degree of similarity between LD and JL conditions. However, consistent with the results from primary tumours, we detected significantly elevated levels of *Cxcl5* in JL BM mononuclear cell samples compared with LD samples (Supplementary Fig. 9e). As CXCL5 has been previously linked to recruitment of suppressive immune cell phenotypes[36], metastatic processes[37–39] and also shown to be under circadian gating[35,40] we assessed its receptor, CXCR2 level in primary tumours. In addition, as a major regulator of tumour cell dissemination we also investigated the expression of CXCR4[41,42], which is also under circadian regulation[35]. In primary tumours, we observed a significant increase in the number of CXCR4$^{+}$ (Fig. 6b), but not of CXCR2$^{+}$ (Supplementary Fig. 9c) cancer cells while the number of CXCR2$^{+}$ TICs was also significantly increased in JL tumours (Fig. 6c). Collectively, these data support the existence of CRD-driven alterations in the cytokine–chemokine network that promote cancer-cell dissemination and the immune-suppressive tumour phenotype: potentially a more prominent CXCL12-CXCR4 axis favouring cancer-cell dissemination and specifically, an enhanced role for the CXCL5-CXCR2 axis that enriches immune-suppressive cell types in the tumour micro-environment. To test this, we explored the effects of CXCR2 on tumour progression in JL mice by treating them with a CXCR2 inhibitor (SB265610)[43,44]. Briefly, 10-week-old MMTV:PyMT mice, who had been subjected to chronic jet lag since the age of 6 weeks, were injected with the CXCR2 inhibitor daily for 5 days then given 2 days' rest; this treatment continued until they reached 18 weeks of age (Fig. 6d). Mice were then sacrificed and examined as in our previous analyses. In the group treated with the CXCR2 inhibitor, we observed a significant decrease in the prevalence of lung metastasis (Fig. 6e) and in the amount of PyMT-positive DCCs in the BM (Fig. 6f). While the percentage of TIC remains similar between conditions (Fig. 6g), the CD4/CD8 ratio was significantly lower in the group treated with the CXCR2 inhibitor, indicating enrichment in cytotoxic CD8 T cells (Fig. 6h, i, Supplementary Fig. 10a). Finally, in agreement with previous studies[37,38,44] we observed that CXCR2 inhibition under LD conditions (Supplementary

Fig. 10b–f) also led to a significant reduction in PyMT-positive DCC and in the CD4/CD8 ratio, without altering the percentage of total TIC, lymphoid and myeloid cells (Supplementary Fig. 10c–f). However, there was no significant difference in the number of metastases detected in LD mice treated with the CXCR2 inhibitor (Supplementary Fig. 10b).

## Discussion

In addition to epidemiological studies, there is a growing body of experimental evidence linking circadian disruption to increased breast cancer risk and poorer survival outcome[6,8,45,46]. Here, we present findings showing that chronic CRD favours cancer-cell dissemination and metastasis formation, and we shed light on the underlying cellular and molecular alterations. For this purpose we used the MMTV::PyMT model of spontaneous murine mammary carcinogenesis[15,47], which recapitulates many processes involved in human metastatic breast cancer[20,48]. This model allowed us to work over extended time periods, representing a physiological context closer to that of human breast cancer, and thus to perform a comprehensive analysis of cancer cells and the tumour microenvironment, with an eye towards systematic changes and metastatic dissemination. Mice were raised under jet-lag conditions that mimicked the effects of shift work or frequent eastbound transmeridian flights; this protocol results in severe perturbations in rest-activity cycles, body temperature and clock-gene expression in the CNS and peripheral organs[17]. It also takes into account the observation that the circadian rhythm is more disturbed by advances rather than delays in local time[49].

Our observations on the systemic physiology of these mice confirmed the importance of circadian rhythm in metabolic processes. In line with previous studies, we found a significant increase in the plasma lipid levels in JL mice compared with controls, supporting the link between CRD and cardiovascular diseases[50,51]. Besides lipid metabolism, the feeding-signalling and insulin-glucose axes are also under circadian regulation. Several studies have reported elevated leptin and insulin resistance in CRD conditions, associated with weight gain, obesity and type-2 diabetes[14,52,53]. In our study, we detected only fine-scale differences in weight and leptin or insulin levels, which might be due to the timeframe of our study and to the continuous feeding activity of JL mice, which reduces physiological differences between rest and activity phases[54]. Furthermore, we did not examine the ability of carcinogenesis to reprogram hepatic homeostasis and metabolism[55,56]. For this reason, we cannot exclude that the developing mammary tumours could have partially rewired

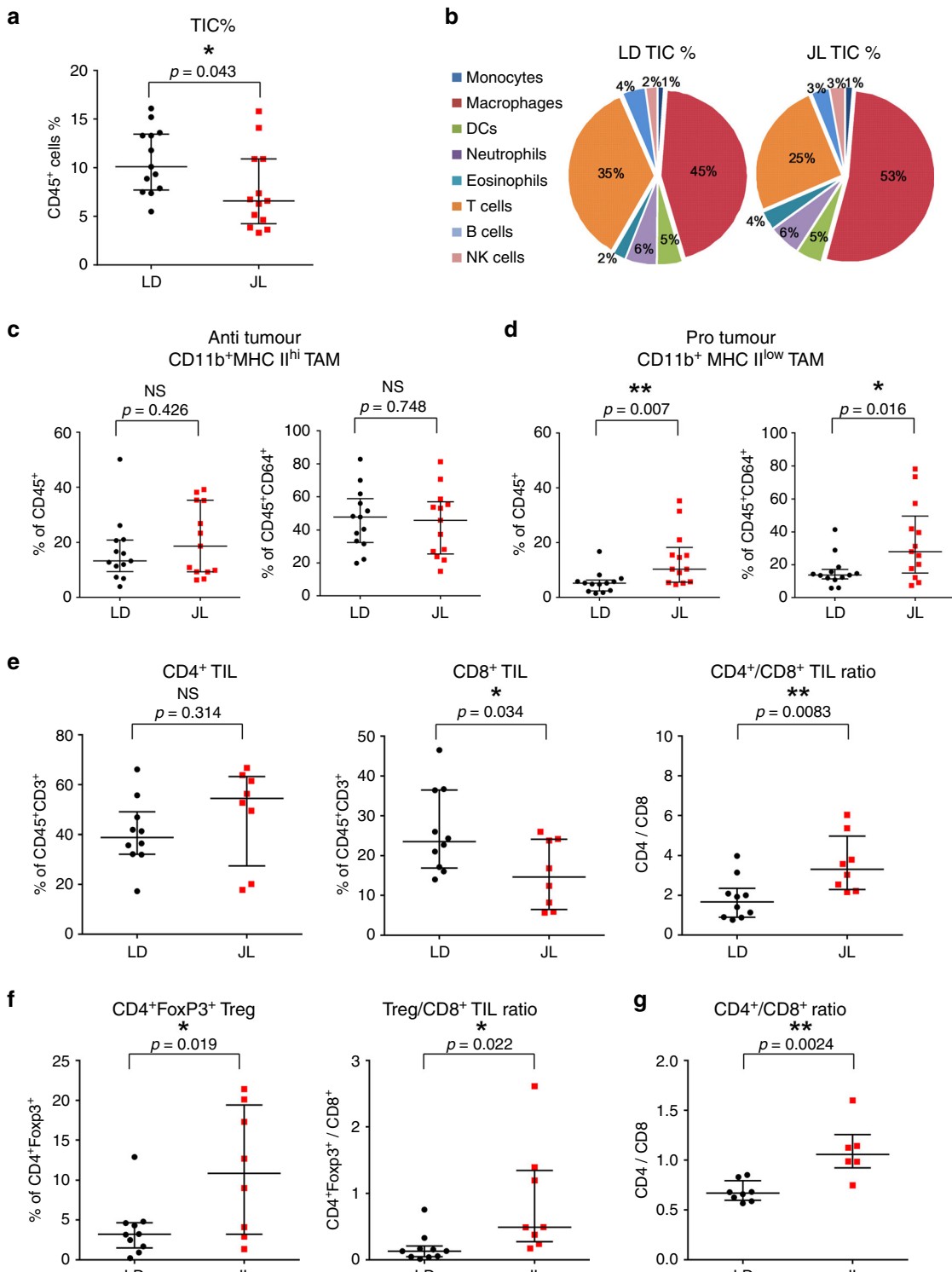

**Fig. 5 Chronic circadian disruption attenuates immune infiltration and creates a pro-tumour immune microenvironment. a** Percentage of tumour-infiltrating immune cells (TIC) in LD ($n = 13$) and JL ($n = 13$) tumours. Flow cytometric analysis revealed a lower percentage of TICs in JL tumours. Data are presented as a scatter dot plot with lines indicating the median with interquartile range (error bars). P-value obtained from an unpaired two-sided t-test. **b** Relative distribution of main immune cell types in TICs from LD and JL tumours. Data presented as pie charts displaying the mean values of 13 mice. **c**, **d** Tumour-associated macrophage (TAM) phenotypes in LD ($n = 13$) and JL ($n = 13$) tumours. Flow cytometry of JL tumours showed a significant reduction in the anti-tumour CD11b⁺MHC II^hi phenotype, with a significant increase in pro-tumour CD11b⁺MHC II^low TAMs. **e** Tumour-infiltrating lymphocytes (TIL) in LD ($n = 10$) and JL ($n = 8$) tumours. TILs are presented as percentage of CD3⁺ immune cells. The proportion of CD8+ TILs was significantly lower in JL tumours, resulting in an increase in the CD4/CD8 ratio. **f** Tumour-infiltrating CD4⁺FoxP3⁺ T cells in LD ($n = 10$) and JL ($n = 8$) mice. Flow cytometry revealed a significant increase of Treg and Treg/CD8⁺ ratio in primary tumours of JL mice. **g** CD4/CD8 ratio in the peripheral blood of LD ($n = 8$) and JL ($n = 6$) mice. **c–g** Data are presented as scatter dot plots with lines indicating the median with interquartile range (error bars). P-values obtained from unpaired two-sided t-test. Indicated (n) represent number of independent experiments as biological replicates.

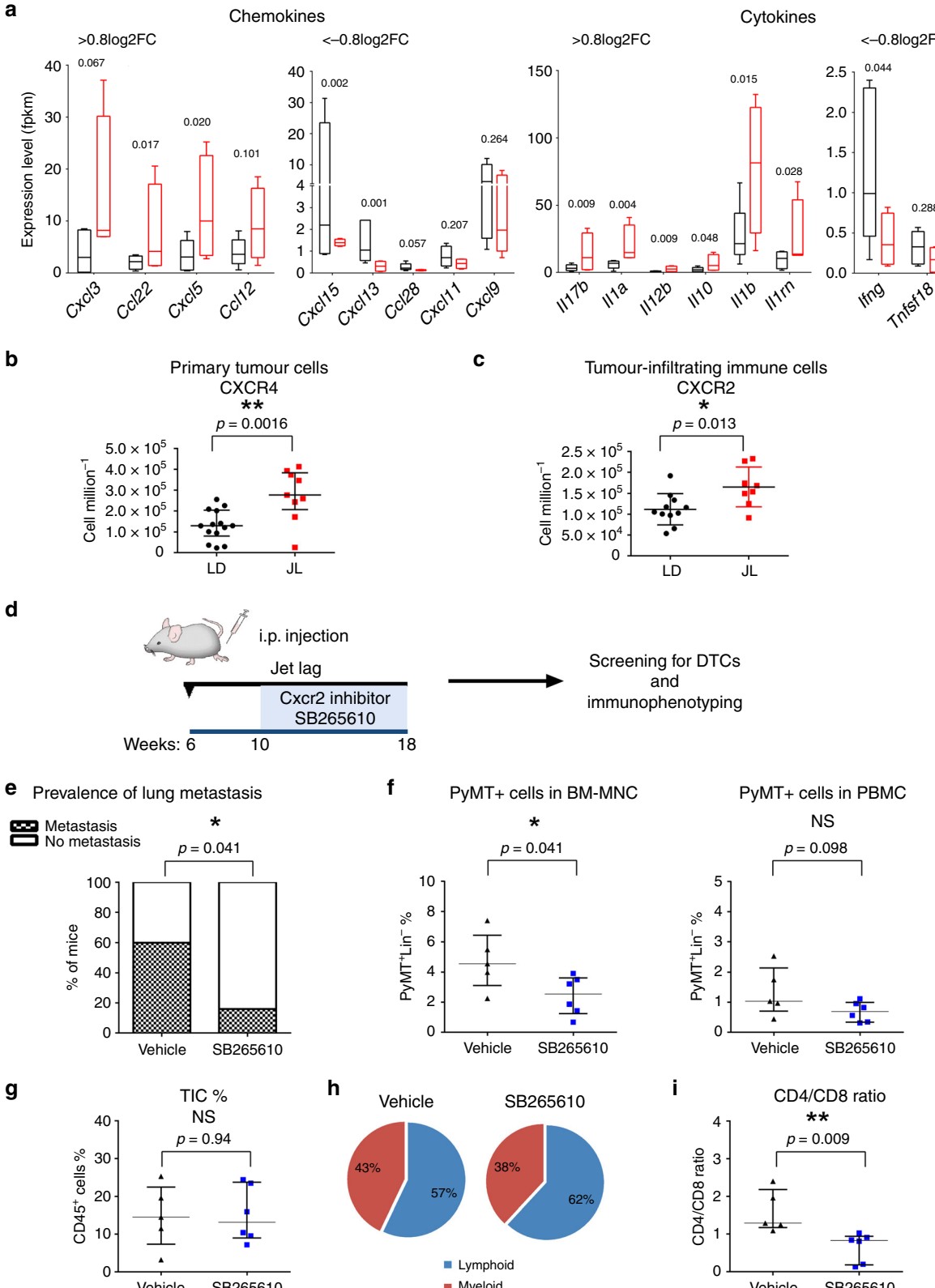

hepatic circadian homeostasis and consequently buffered the CRD-induced metabolic changes.

Consistent with the slight metabolic changes observed between conditions, we did not observe major histological differences in primary tumours following 10 weeks of CRD. This supports the hypothesis that the increase in DCCs and the observed enrichment in malignant lesions in JL mice represent a global speed-up toward carcinogenesis rather than a selective process leading to the development of different tumour subtypes in LD and JL mice.

Previous studies proposed that CRD boosts tumour progression through increased proliferation and metabolic reprogramming[17,56–59]. Here we demonstrate that CRD also leads to significantly enhanced cancer-cell dissemination and metastatic burden. Our results provide experimental evidence that reinforces

**Fig. 6 Circadian disruption alters chemokine/cytokine regulatory networks. a** The most up- and downregulated chemokines and cytokines in LD ($n = 5$) and JL ($n = 4$) tumours, as detected by mRNA-seq. Expression data represent FPKM values and are presented as box-and-whisker plots. Variability is depicted using medians (line in the box), 25th and 75th percentiles (box), and min to max (whiskers). P-values were calculated using DESeq2 and the Wald test based on the negative binomial distribution. Respective log2FoldChange (log2FC) are listed in Supplementary Data 3. **b** Number of tumour cells positive for CXCR4 in LD ($n = 14$) and JL ($n = 9$) tumours. Data are shown as a scatter dot plot with lines indicating the median with interquartile range (error bars). P-values obtained from unpaired two-sided t-test. **c** Number of tumour-infiltrating immune cells positive for CXCR2 in LD ($n = 11$) and JL ($n = 8$) samples. Data are shown as a scatter dot plot with lines indicating the median with interquartile range (error bars). P-values obtained from unpaired two-sided t-test. **d–i** Effects of CXCR2 inhibition on tumour development in JL mice. **d** Scheme illustrating the experimental plan. MMTV:PyMT mice were subjected to chronic jet lag from the age of 6 weeks, and from 10 to 18 weeks of age were intraperitoneally (i.p.) injected once a day (5 days injection + 2 days resting) with the CXCR2 antagonist SB265610 (2 mg kg$^{-1}$ in 5% DMSO–8% Tween80 0.9% NaCl). Mice were sacrificed at 18 weeks of age. **e** Prevalence of lung metastasis in mice injected with vehicle ($n = 5$) or SB265610 ($n = 6$). P-value obtained from a binomial two-sided test. **f** The percentage of disseminated tumour cells in BM and peripheral blood in mice injected with vehicle ($n = 5$) and SB265610 ($n = 6$). **g** Percentage of tumour-infiltrating immune cells (TIC) in mice injected with vehicle ($n = 5$) and SB265610 ($n = 6$). **h** Relative distribution of lymphoid and myeloid TICs in both cohorts. Data are presented as pie charts displaying the mean values of mice. **i** CD4/CD8 ratio in tumours from vehicle ($n = 5$) and SB265640 ($n = 6$) groups. Data (**f**, **g**, **i**) are presented as scatter dot plots with lines indicating the median with interquartile range (error bars). P-values were calculated from an unpaired two-sided t-test. Indicated ($n$) represent number of independent experiments as biological replicates.

the findings of previous genetic studies linking cancer severity, relapse and higher risk of metastasis with the compromised expression of clock genes[8,60,61]. Specifically, we observed reduced expression of the *Per2* and *Cry2* clock genes in primary tumours of JL mice. Intriguingly, genes involved in phototransduction were among the most significantly altered in both primary tumours and mononuclear BM cells from JL mice. The expression of several phototransduction genes is under circadian regulation through the NR1D1 (Rev-Erbα)—NRL, CRX and NR2E3 complexes[62,63], and their downregulation is certainly related to the CRD conditions experienced, despite the fact that the analysed tissues were not directly exposed to light stimuli. In mammals, the functionality of phototransduction molecules in non-visual tissues has only been poorly investigated. Some data have suggested a light-independent mechanism of activation of the photo-transduction (transducin/PDE6/Ca$^{2+}$/cGMP) cascade through Wnt/Frizzled-2, which might function as an anti-apoptotic mechanism[64,65]. Two recent studies also showed that peripheral clocks can be synchronised by light independently of a functional central circadian clock, and suggested that phototransduction players could drive this[66,67].

Clock genes and circadian oscillation have been shown to regulate the EMT programme[13,68], which is one of the mechanisms behind the spread and colonisation of tumour cells[23,69]. Our differential gene expression analysis of primary tumours revealed an elevated expression of genes associated with EMT under CRD conditions. The upregulated EMT-inducers—*Zeb2*, *Foxc2* and *Inhba*—have also been implicated in promoting cancer stem-cell and metastatic properties[70–72]. As found in previous studies, our data support the link between the EMT process and the stemness/tumour-initiating ability of cancer cells[21,22,73]. In particular, we observed enrichment in the stem-cell population of primary tumours from JL mice. Furthermore, our data reveal an enhanced degree of stemness and tumour-initiation potential for JL cancer cells in vitro and in vivo. In addition, we demonstrate here that the stemness of mammary epithelial cells oscillates diurnally. This provides further evidence for the role that circadian rhythm and clock genes play in regulating the stemness of mammary cells. The negative correlation between oscillatory *PER2* expression and capacity for mammosphere formation confirms the suppressive effect of *Per2* on self-renewal and tumour initiation[13]. Similarly, mammary epithelial cells derived from ClockΔ19 mutant mice showed reduced mammosphere-formation capacity[12]. Collectively, these data suggest that CRD promotes metastasis by enhancing the EMT programme and, consequently, the tumour-initiating potential of cancer cells.

Circadian regulation of leukocyte homeostasis, trafficking and immune responses, especially in contexts of infection and inflammation, is well-documented[74–76]. However, little is known about the effect of circadian rhythms on the tumour micro-environment and tumour immunity. Our results demonstrate that CRD has a profound effect on tumour immunity through modulation of the cytokine–chemokine network. Our data suggest that CRD attenuates immune infiltration to tumour sites potentially by disrupting the diurnal trafficking of leukocytes and consequently reducing the daily total number of these cells in the circulation[77]. Here, we provide evidence that CRD induces a pro-tumourigenic switch of the tumour immune microenvironment, primarily driven by alterations in the CXCL5-CXCR2 axis. The circadian expression of the CXCL5-CXCR2 axis has been described in previous studies, along with its involvement in inflammatory diseases[35,40,78,79]. Furthermore, CXCR2 and several of its ligands (CXCL1, CXCL2, CXCL5, CXCL7 and CXCL8) have also been linked to breast cancer progression and metastatic invasion[80]. Therefore, considering the results of previous studies, we propose the following inflammatory cascade as a possible mechanism behind CRD-related enhanced tumourigenesis and metastatic spread (Fig. 7): CRD increases the expression of *Cxcl5* in tumours, leading to enhanced infiltration of CXCR2$^+$ myeloid cells, e.g. MDSCs. The consequent accumulation of MDSCs, TAMs and TANs promotes an immunosuppressive micro-environment[36]. These cells are able to directly suppress T-cell responses and inhibit CD8 T-cell infiltration, resulting in impaired anti-tumour activity[36,81–83]. Collectively, this autocrine cascade promotes tumour growth and metastasis. In JL mice, the inhibition of CXCR2 significantly reduced lung metastasis and dissemination to bones, which strongly supports our hypothesized model of CRD-linked tumourigenesis. Two mechanisms have been described that explain how the inhibition of CXCR2 signalling suppresses metastasis. First, CXCR2 signalling has been shown to play a role in the modulation of the tumour immune microenvironment and the recruitment of MDSCs in distant metastatic sites, along with the consequent development of the pre-metastatic niche[37,38]. Second, the CXCL5-CXCR2 axis has also been associated with the process by which circulating tumour cells home to the bone[39]. In addition, a more recent study showed its importance in colonization during bone metastasis[84]. In line with these studies, we observed fewer cancer cells in the BM after inhibition of CXCR2 in both LD and JL conditions, and a lower degree of metastasis in JL condition (but not in the LD group), but it remains unclear which one of these mechanisms (or both) might be responsible. Here, in line with previous studies[37,38], we suggest the use of CXCR2 inhibition in combination

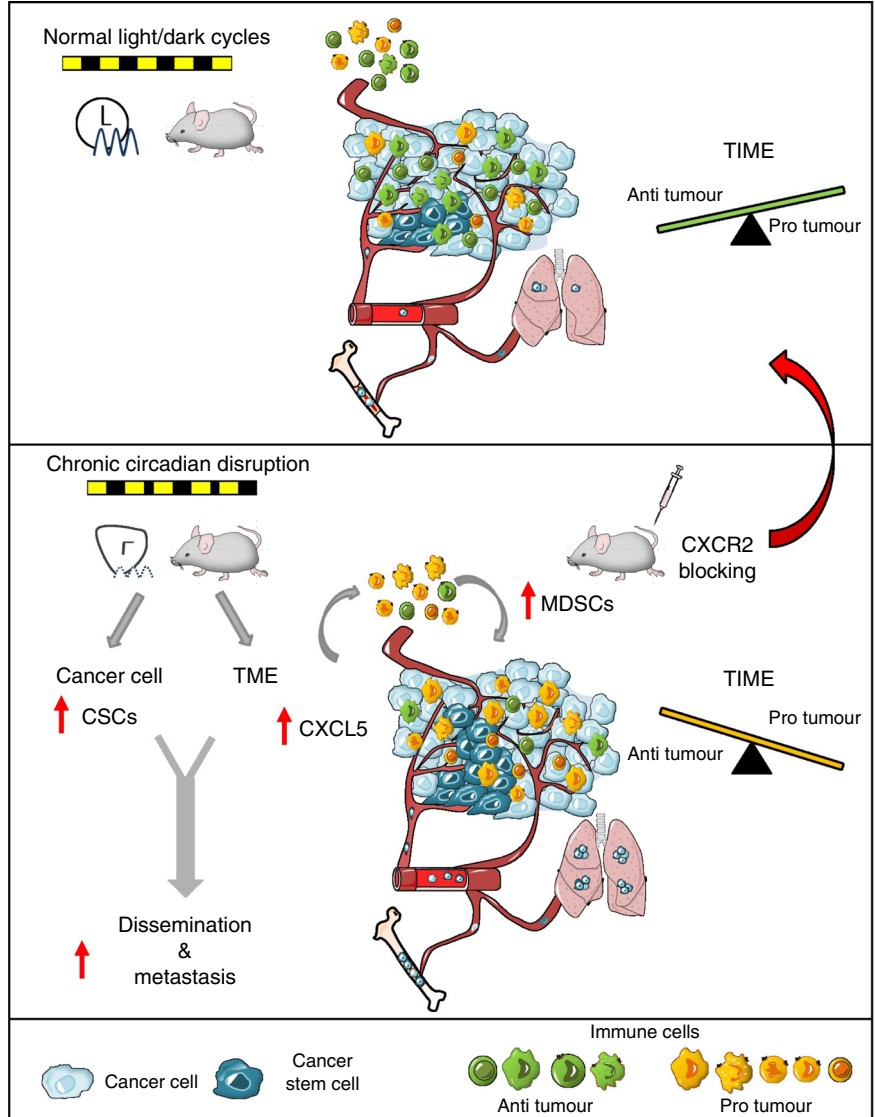

**Fig. 7 Conceptual schema.** Chronic circadian disruption (CRD) alters two key aspects of tumour biology. It increases the proportion of cancer stem cells (CSCs, in dark blue) and modifies the tumour microenvironment (TME) through the recruitment of myeloid-derived suppressor cells (MDSCs, in yellow), resulting in a suppressive tumour immune microenvironment (TIME). At least one mechanism that could drive this process is an enhanced CXCL5-CXCR2 axis in the TIME. Collectively, these effects result in increased dissemination and metastasis in bone marrow and lungs. Inhibition of the CXCR2 axis is able to alleviate the effect of CRD and recover anti-tumour activity.

with conventional chemotherapy or immunotherapy to reduce cancer-cell dissemination and improve therapy outcome. This could be especially beneficial in cases with known CRD (either systemic or localised, tumour specific alteration of the circadian molecular clock) where the CXCR2 driven mechanisms are accelerated, as our results show.

Furthermore, we also describe here upregulation in the CXCL12-CXCR4 axis, which is another key mechanism that promotes metastatic spread in JL mice. Circadian control of the CXCL12-CXCR4 axis[35,77,85] and its role in immunosuppression and breast cancer metastasis are well described[41,42,86].

The CRD-induced alterations we describe in chemokine/chemokine-receptor signalling drive the changes in the tumour microenvironment and metastatic capacity. Recent studies have also suggested the use of CXCR2 or CXCR4 inhibitors in combination with immunotherapy to overcome therapeutic resistance[83,87]. These therapeutic strategies, together with the use of predictive tools based on clock-gene

expression for cancer prognosis, will lead to precision circadian medicine with improved efficacy and responsiveness to immunotherapy[46,88].

In conclusion, our study provides, experimental evidence of the link between CRD and an increase in cancer-cell dissemination and metastasis. Based on our results, we propose two molecular mechanisms by which CRD promotes tumourigenesis: (1) CRD increases the stemness and tumour-initiating potential of cancer cells, at a minimum by promoting a pro-EMT intra-tumoural context, and (2) CRD reduces anti-tumour immunity through modifications in chemokine/chemokine-receptor signalling. Finally, our findings draw attention to the potential role of genes involved in phototransduction in peripheral tissues. However, our data do not confirm whether the observed downregulation of these genes in JL mice is due to their circadian rhythmicity in LD mice or whether it reflects a functional role in peripheral tissues. This intriguing observation in bone marrow cells and primary tumours during CRD need further investigation.

## Methods

**Mouse and tumour model.** Mouse strains MMTV-PyMT (FVB/N background, JAX #002374) and MMTV-Luc2 (B6; FVB background, JAX #027827) were purchased from the Jackson Laboratory. Five-week-old female C57BL/6JOlaHsd mice were purchased from Envigo (The Netherlands).

To establish our experimental colony, the FVB PyMT mice were bred to C57Bl/6J (B6) mice for three generations. Mice in the F4 generation of the B6*FVB PyMT cross were then bred to B6*FVB MMTV-Luc2 mice. The F1 generations of these PyMT and Luc2 crosses were used in our experiment. We observed that these mixed-background mice (B6*FVB PyMT::Luc2) developed palpable mammary tumours by 10–12 weeks of age and reached the late/exponential phase of tumour growth at 18–20 weeks of age. Pulmonary macro-metastases were detected starting from 16 weeks, with increased prevalence at 19–20 weeks of age. Based on these observations, we decided to sacrifice mice at 16 weeks of age (Fig. 1a), in the early/mid phase of tumour growth. Details of tumour evaluation and sample processing can be found below.

Mice were crossed, reared and housed in the institutional animal facility SEIVIL (Service D'Experimentation Animale in vivo INSERM Lavoisier). Mice were housed in groups in standard rectangular cages with unrestricted access to food and water and provided with nesting material. Mean temperature and humidity were 22 °C and 50%, respectively. Telemetry experiments were performed in the LET (Laboratoire d'Enregistrement Télémétrique, Inserm U776). All mouse experiments were performed in accordance with DIRECTIVE 2010/63/EU guidelines and were approved by the CEEA26-Paris-Sud Ethics Committee and the French Ministry of Higher Education and Research (APAFIS #8475-2017011015 222063, APAFIS #13183-2018012412194882, APAFIS #22201-2019093016134321).

**Cell-line culture.** The immortalised non-tumourigenic human epithelial cell line MCF12A (ATCC, CRL-10782)[89] was cultured in DMEM/F12 (Gibco) supplemented with 5% horse serum (Sigma), 10 ng ml$^{-1}$ cholera toxin (Sigma), 10 µg ml$^{-1}$ insulin (Sigma), 0.5 µg ml$^{-1}$ hydrocortisone (Sigma), 20 ng ml$^{-1}$ human recombinant epidermal growth factor (StemCell Technologies), 1% Penicillin/Streptomycin (Gibco) and 1% L-Glutamine (Lonza).

**Jet-lag conditions.** Mice were kept under normal 12/12 h light/dark (LD) conditions until 6 weeks of age. At this point they were randomly assigned either to remain in LD or to be exposed to jet-lag (JL) conditions, with an 8-h advance in the light/dark cycle every 2 days[17,59]. To induce chronic circadian rhythm disruption (CRD), mice were kept in JL conditions for 10 weeks in a specialised chronobiological facility; all mice had free access to food and water. Zeitgeber time (ZT) 0 corresponded to the onset of light, while ZT12 corresponded to the onset of dark. LD mice were sampled at ZT3–ZT4; in the case of JL mice, ZT was not followed.

**Rest/activity and temperature recording.** We used the DSI Implantable Telemetry system (Data Sciences International, St. Paul, MN) to assess locomotor activity and core body temperature of mice kept in either LD or JL conditions. A telemetric transmitter (PhysioTel TA-F10, DSI) was implanted in the peritoneal cavity of each mouse under isoflurane anaesthesia. Dataquest A.R.T. v4.30 software was used for data collection. Data were recorded every 10 min throughout the experiment. We implanted transmitters in 6 mice at the age of 14 weeks (mice had reached the required weight of 20 g). Mice were kept 2 weeks in standard light conditions (LD) for recovery/synchronisation, then they were randomly assigned to either the LD or JL protocol for 4 weeks of recording. Actograms were created using Microsoft Excel following the method described by Oike et al.[90]. Actogram data were analysed and visualised in ImageJ using the ActogramJ software package[91] and in GraphPad Prism v601. Data from D10–D20 (mice age of 15 to 16 weeks) and D32–D46 (mice age of 18 to 19 weeks) post-implantation were used for periodogram analysis. Period was determined in ActogramJ software package for activity and core body temperature patterns using Fourier and Lomb-Scargle methods, respectively. To confirm circadian rhythmicity of activity and temperature patterns we defined the coefficients of determination ($R^2$) by fitting a cosine algorithm in GraphPad Prism v601[29].

**Bioluminescence imaging.** For in vivo imaging of tumours, mice were intraperitoneally (i.p.) injected with 150 mg luciferin kg$^{-1}$. Luciferin was resuspended in PBS at a concentration of 15 mg/ml and filter-sterilised with a 0.22-µm filter. Following injection, mice were anaesthetised with isoflurane gas (2% during imaging period). Bioluminescence imaging was performed 15, 20, 25 and 30 mins after injection, as kinetic analysis determined that peak luciferase activity occurred 20–25 min post-injection (IVIS Spectrum imaging system, PerkinElmer). Images were analysed with Living Image software. To determine tumour burden, total flux (photons per second) was measured in a fixed region of interest (whole body).

**Sample collection.** Mice were anesthetised with isoflurane inhalation. Blood was collected by cardiac puncture and placed in EDTA (10% 0.5 M EDTA). Following cervical dislocation, we collected tumours/mammary glands, lungs, and both hindlimb bones and stored them in ice-cold PBS until further processing. LD mice were sampled at ZT3–ZT4; in the case of JL mice, ZT was not followed.

A blood cell count analysis was performed on 60 µl of whole blood using a VETSCAN HM5 haematology analyser (ABAXIS). The remainder of the blood was spun down at $300 \times g$ for 20 min at 4 °C; from this, 150–300 µl plasma were collected and stored at −80 °C until further analysis. Erythrocytes were lysed using 1X red blood cell (RBC) lysis buffer (15.5 mM ammonium chloride, 1 mM potassium bicarbonate, 10 µM EDTA in distilled water). White blood cells were frozen in 10% DMSO/FBS and preserved at −150 °C for flow cytometry analysis.

Tumours were weighed to calculate tumour burden (tumour burden % = [tumour weight (g) per body weight (g)] × 100). A small portion of the tumours was fixed in 4% paraformaldehyde (PFA) for histological analyses. The rest of the tumours was dissociated (see details below).

Lungs were incubated in 1 µg ml$^{-1}$ heparin in PBS for 1 h at 4 °C, then fixed in 4% PFA for 2–3 days.

Bone marrow (BM) was isolated from femurs and tibias of either one or both hind limbs. BM cells were washed two times with PBS. Erythrocytes were removed by red blood cell lysis. BM cells were frozen in 10% DMSO/FBS and preserved at −150 °C for flow cytometry analysis. In the cases in which only one hindlimb was flushed out, the other was fixed in 4% PFA for histological examination.

**Tumour dissociation and cell isolation.** Tumours were minced finely with scalpels. Samples were placed into 50-ml tubes with 10–15 ml of pre-warmed dissociation media (1 mg ml$^{-1}$ collagenase type I, 0.5 mg ml$^{-1}$ Dnase I and 1% Penicillin-Streptomycin in DMEM F12) and incubated at 37 °C on a rotator shaker for 1 h. Samples were gently vortexed every 20 min to avoid clumping. Following incubation, samples were filtered through a 70-µm nylon strainer and topped up to 40 ml with PBS/EDTA (2 mM). Cells were spun down at $450 \times g$ for 10 min. The top fatty layer was collected, topped up to 40 ml with PBS/EDTA, gently vortexed, and centrifuged. Tumour cells were resuspended in 2 ml 1x RBC lysis buffer and incubated for 5 min to remove red blood cells. The cell suspension was filtered through a 40-µm nylon strainer. Single cells were frozen in 10% DMSO/FBS and preserved at −150 °C for flow cytometry analysis and tumour-initiation study.

**Detection and quantification of lung metastasis.** Lungs fixed in 4% PFA were washed in PBS then permeabilised with 0.1% Triton X/PBS for 2 days. Lungs were incubated in 3% hydrogen peroxide ($H_2O_2$) for 30 min, then the solution was diluted to 1% $H_2O_2$ with PBS and incubated overnight. Blocking was performed with 5% FBS/1% BSA/PBS for 24 h. Lungs were again rinsed with PBS, then incubated with anti-PyMT antibody (sc-53481, Santa Cruz, 1:100 dilution) for 48 h. Following overnight washing, lungs were incubated with rabbit anti-rat IgG AP (Thermo Fisher, 1:1000 dilution) secondary antibody for 24 h. To reveal antibody labelling, lungs were placed into NBT/BCIP working solution. After colour developed, lungs were washed, each lobe was imaged using a stereomicroscope, and metastatic foci were counted. All incubations and washing were performed at 4 °C.

**Bone tissue processing and analysis.** Tissues samples were fixed in 4% paraformaldehyde overnight at 4 °C. Micro-CT imaging and analyses were performed with a SkyScan 1172 apparatus (Bruker micro-CT), using a voxel size of 12 µm, a voltage of 50 kV, an intensity of 200 µA and an exposure of 900 ms. NRecon reconstruction and DataViewer software were used for 3D reconstruction. Analysis of structural parameters was performed using CTVox. After the micro-CT scan, hind limbs were processed to obtain paraffin sections for histological analysis as follows: Hind limbs were decalcified in 20% EDTA (pH 7.5) at 4 °C with constant shaking for 15 days; the EDTA solution was replaced every 2–3 days. Decalcified bones were embedded in paraffin and sectioned at 5 µm using a Leica microtome. Sections were stained with hematoxylin and eosin.

**Flow cytometry.** We characterised tumour cells using the following antibodies: CD45 PB (1:200), CD31 PB (1:500), TER-119 PB (1:200), CD140A BV421 (1:100), CD24 BV510 (1:100), CD29 FITC (1:100), CD49f PE (1:100), CD44 Pe/Cy7 (1:100), CD90.1 APC (1:100), CD326 APC/Vio770 (1:100), CD45 PE/Dazzle594 (1:100), CXCR1(CD181) AF750 (1:50), CXCR2(CD182) PE/Vio770 (1:50), and CXCR4(CD184) APC (1:50). Tumour cells were identified as Lin$^-$ (CD45$^-$CD31$^-$CD140a$^-$Ter119$^-$). To phenotype tumour-infiltrating immune cells we used the antibodies CD45 PB (1:200), CD24 BV510 (1:200), CD11b FITC (1:100), CD64 PE (1:200), CD11c Pe/Cy7(1:100), MHC II (IA/AE) APC (1:1500) and Ly6G APC/Vio770 (1:100), and a gating strategy as described by Yu et al.[30] and shown in Supplementary Fig. 7A. Major phenotypes of tumour-infiltrating lymphocytes (TILs) and peripheral blood T cells were identified using the antibodies CD45 PE/Dazzle594 (1:50), CD3 BV510 (1:20), CD8a APC/Fire750 (1:50) and CD4 AF488 (1:50). To detect CD4+FoxP3+ regulatory T cells we used the True-Nuclear Mouse Treg Flow kit (FoxP3 AF488 (1:20), CD4 APC/CD25 PE (1:10), Biolegend) in combination with CD45 PE/Dazzle594 (1:50), CD3 BV510 (1:20) and CD8a APC/Fire750 (1:50). Gating strategy shown in Supplementary Fig. 7b.

To detect disseminated tumour cells, mononuclear cells from blood or bone marrow were, respectively, stained for CD45 (1:100) or CD45 (1:100), CD31 (1:500), TER-119 (1:200) and CD140A (1:100). Following cell-surface staining, cells were fixed with 1% PFA/PBS for 30 min then permeabilised with 0.1% Triton X/ PBS for 20 min. Cells were blocked with 5% FBS/1% BSA/PBS for 30 min.

Following a PBS wash, cells were incubated with anti-PyMT AF488 antibody (1:10) for 1 h. All incubations and staining were performed at 4 °C.

To distinguish between live/dead cells, either propidium iodide (PI, 1:1000) or Zombie Violet (1:500) fixable viability stain was used. Isotype controls were used, respectively, at corresponding concentration. Detailed antibody list provided as Supplementary Table 4.

FlowJo Software (version 10, Tree Star Inc) was used for flow cytometry data analysis.

**RNA isolation and RT-PCR.** Total RNA was extracted from bulk of dissociated cells from primary tumours and from bulk of bone marrow mononuclear cells using the PureLink RNA kit (Invitrogen) according to the manufacturer's instructions. cDNA synthesis was performed using the High-Capacity cDNA Reverse Transcription kit (Applied Biosystems) with Oligo(dT) primers (ThermoScientific). RT-PCR was performed with QuantStudio 5 (384-well format, Thermo Fisher) using FastStart Universal SYBR Green Master mix (Roche). Samples were analysed in replicates and melting curve analysis was performed for each run. The geometric mean of Ct values for *Ctbp1*, *Prdx1* and *Tbp* or *36B4*, *RPL4*, *HSPCB* and *TBP* were, respectively, used for normalisation of mice and human samples. Relative fold change ($2^{-\Delta Ct}$) and gene expression were represented in arbitrary units. Primer sequences were:

*PyMT* fw: CTCCAACAGATACACCCGCACATACT, rv: GCTGGTCTTGG TCGCTTTCTGGATAC

*Cxcl5* fw: GGGAAACCATTGTCCCTGA, rv: TCCGATAGTGTGACAGA TAGGAAA

*Cxcl3* fw: CAGCCACACTCCAGCCTA, rv: CACAACAGCCCCTGTAGC

*Il1b* fw: GCTTCCTTGTGCAAGTGTCT, rv: GGTGGCATTTCACAGTTGAG

*Ctbp1* fw: GTGCCCTGATGTACCATACCA, rv: GCCAATTCGGACGATGA TTCTA

*Prdx1* fw: AATGCAAAAATTGGGTATCCTGC, rv: CGTGGGACACACAAA AGTAAAGT

*Tbp* fw: AGAACAATCCAGACTAGCAGCA, rv: GGGAACTTCACATCA CAGCTC

*BMAL1* fw: TGGAGAAGGTGGCCCAAAGA, rv: TCCTCAGCAATCATT CGGCCTA

*PER2* fw: GTCCACCTCCCTGCAGACAA, rv: CTGGTAATACTCTTCA TTGGCTTTCA

*36B4* fw: GTGATGTGCAGCTGATCAAGACT, rv: GAAGACCAGCCCAAA GGAGA

*RPL4* fw:GCCAGGAATCACAAGCTCCG, rv: CCGCCGCCTTCTCATCT GAT

*HSPCB* fw: AGAAGGTTGAGAAGGTGACAATC, rv: AGTTGTCCCGAAGT GCCTG

*TBP* fw: TGCTGCGGTAATCATGAGGA, rv: GTCTGGACTGTTCTTCAC TCTT.

**mRNA-seq analysis.** mRNA-seq was performed by BGI (Hong Kong) using their standard procedures. mRNAs were purified using oligo(dT)-attached magnetic beads. cDNA was synthesised from fragmented mRNAs using random hexamer-primers, then the synthesised cDNA was subjected to end-repair and 3′ adenylation. Adapters were ligated to the ends of these 3′ adenylated cDNA fragments, and cDNA fragments were then amplified by PCR and purified with Ampure XP Beads (AGENCOURT). The size and quantity of libraries were validated on an Agilent 2100 Bioanalyzer. The double-stranded PCR products were heat-denaturated and circularised using a splint oligo sequence. The single-strand circular DNA (ssCirDNA) was formatted as the final library. Each library was amplified with phi29 to make a DNA nanoball (DNB) which contained more than 300 copies of the given molecule. The DNBs were loaded into the patterned nanoarray and single-end 50-base reads were generated on a BGISEQ-500 sequencing platform. For each library a minimum of 23 million single-end reads were produced. All raw sequencing reads were filtered using SOAPnuke software (https://github.com/BGI-flexlab/SOAPnuke) to remove reads with adaptors, reads made up of more than 10% unknown bases, and low-quality reads. The resulting clean reads were then stored in FASTQ format. Reads were mapped using Bowtie2[92] on the mouse reference genome mm10 (GRCm38); gene expression levels were calculated with RSEM[93] and normalised using the value of fragments per kilobase of exon per million fragments mapped (FPKM) in each sample. Normalised gene expression values were used to produce heatmaps using the ComplexHeatmap package of R[94], using Euclidean distances and the agglomeration method Ward.D2 of the function hclust. Principal Components Analyses were performed using the pca function of the mixOmics package[95]. Differentially expressed genes were detected using DESeq2, with un-normalised counts as input[96]. DEGs were then used to assess pathway functional enrichment with the KEGG annotation database, using the R package hypeR[97]. DEGs were ranked in a list according to their p-values, and Gene Set Enrichment Analysis (GSEA) was performed using GSEA software from the Broad Institute[98].

**Immunohistochemistry.** Primary tumours (fixed in 4% PFA) were washed in PBS and transferred to 70% EtOH. They were then embedded in paraffin wax according

to standard histological protocols. Paraffin-embedded-sections (5-µm-thick) were mounted on adhesive slides (Klinipath-KP-PRINTER ADHESIVES), then deparaffinised and stained with HES (hematoxylin, eosin and saffron) for morphological observation of tumours. All slides were scanned with a Pannoramic Scan 150 (3D Histech) and analysed with a CaseCenter 2.9 viewer (3D Histech).

**Tumour-initiation study.** Cancer cells were enriched by magnetic bead-based negative selection using a cocktail of CD45, CD140a, CD31 and Ter119 biotinylated antibodies (Miltenyi Biotec), followed by labelling with streptavidin microbeads. Flow-through cells were collected, counted, and kept on ice until further use. One-hundred thousand live tumour cells from each donor were injected into the right 4th mammary fat pad of 10-week-old wild-type C57BL/6J mice. In preparation for injection, tumour cells were resuspended at a concentration of $5 \times 10^6$ cells/ml in PBS. Twenty microliters of cell suspension was mixed with 20 µl of Geltrex (Gibco) and immediately injected into the mammary fat pad using a U-100 insulin syringe. Host mice were kept in LD conditions from 6 weeks of age until sacrifice at 18 weeks.

**CXCR2 inhibition.** Mice were kept under JL conditions starting from 6 weeks of age; inhibition treatment took place from 10 to 18 weeks of age. The CXCR2 antagonist SB265610 was resuspended in 5% DMSO–8% Tween80 in 0.9% NaCl and injected into mice daily (5 days i.p. injection + 2 days resting) at a concentration of 2 mg kg$^{-1}$. The same experiment was performed on LD mice, for which the injection time corresponded to ZT6-ZT7, when the highest number of CD45+ cells was detected in the peripheral blood[82]. Under jet-lag conditions no specific ZT was kept, but injections were always performed at the same time of the day (between 1 and 2 pm, at the same time as the LD group). A subset of each group was injected only with vehicle and used as control. JL mice were sacrificed at 18 weeks of age. In LD conditions, however, this experiment was performed on PyMT- FVB* Luc2-B6 F1 hybrid mice, which exhibited faster tumour progression. For this reason, LD mice were sacrificed at the age of 14 weeks.

**Plasma chemistry and endocrine analysis.** All measurements were performed by the Clinical Chemistry and Haematology platform of Phenomin-ICS (Institut Clinique de la Souris, Strasbourg). Blood chemistry (glucose, albumin, CK, LDH, ASAT, ALAT, ALP, a-amylase, total cholesterol, HDL and LDL cholesterol, triglycerides and creatinine) was analysed on an OLYMPUS AU-480 automated laboratory work station (Beckmann Coulter, USA) with the kits and controls supplied by Beckmann Coulter. Amounts of free fatty acids were measured on the AU-480 using a kit from Wako (Wako Chemical Inc, Richmond, USA). Internal quality control materials (Olympus) were analysed on a daily basis to ensure precision throughout the experiment. Insulin levels were measured on a BioPlex analyser (BioRad) using the Mouse Metabolic Magnetic bead panel kit (Reference: MMHMAG-44K—Milliplex map by Millipore). Corticosterone was measured by RIA using the Corticosterone 125I RIA kit for rats & mice (MP biomedical: 07-120102)

**Luminex assay.** A mouse magnetic multiplex Luminex assay was purchased from R&D Systems-biotechne for 18 analytes: MCP-1/CCL2, KC/CXCL1, MIP-2/CXCL2, LIX/CXCL5, SDF1/CXCL12, G-CSF, GM-CSF, IFNγ, IL-1β, IL-2, IL-4, IL-6, IL-10, IL-12p70, Leptin, M-CSF, TNFα and VEGF. Measurements were performed by the Cochin Cytometry and Immunobiology Facility (CYBIO, Institut Cochin, Paris) using a Bioplex 200 apparatus (Luminex).

**Circadian synchronisation of cells.** MCF12A cell were entrained by serum shock. Cells were seeded into 60-mm dishes at a density of 200,000 cells per dish. Following 4 days of culture in complete medium, cells were washed with PBS and starved for 12 h in basal DMEM/F12 medium. Cells were then stimulated with serum shock (complete growth medium supplemented with 50% of horse serum) for 2 h. At the end of synchronisation (Zeitgeber Time ZT0), cells were washed with pre-warmed PBS and placed in pre-warmed DMEM/F12 that was supplemented with 20 ng ml$^{-1}$ hrEGF. Cells were collected at ZT24, ZT36, ZT48 and ZT60 for the mammosphere-formation assay.

**Mammosphere formation.** To examine mammosphere formation, we plated either 500 primary murine tumour cells or 1000 MCF12A cells per well into ultra-low-attachment 24-well plates. Cells were cultured in DMEM/F12 that was supplemented with 5 µg ml$^{-1}$ insulin, 4 µg ml$^{-1}$ heparin, 5 µg ml$^{-1}$ hydrocortisone and 20 ng ml$^{-1}$ recombinant murine or human EGF, respectively. Mammosphere formation was assessed in 2–4 replicates. Spheres were imaged after 20 days for primary tumour cells and 8 days for MCF12A. Measurements were performed in ImageJ software; the threshold was 50 µm for primary tumourspheres and 40 µm for normal human spheres. Mean values of replicates were used for data analysis. Mammosphere-formation efficiency (MFE) was calculated and presented as a percentage.

**Statistical analysis.** GraphPad Prism v6.01 (GraphPad Software, USA) was used to prepare all graphs and perform statistics, unless stated otherwise. Results were

analysed using *t*-tests (unpaired, two-sided), binomial two-sided test or one-way ANOVA, as appropriate.

## Data availability

The RNA sequencing data are available at the Sequence Read Archive (https://www.ncbi.nlm.nih.gov/sra) under the study accession number PRJEB33802 . KEGG annotation database is available online at https://www.genome.jp/kegg/annotation/. All other relevant data are available in the Article, Supplementary Information or from the corresponding authors upon reasonable request. The source data underlying Figs. 1b–f, 2a–d, 3b–f, 4a–d, f, 5a–g and 6a–c and e–i, and Supplementary Figs. 1, 2, 6, 8–10 are provided as a Source data file. A reporting summary for this article is available as a Supplementary Information file.

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

## Acknowledgements

We thank all members of UMRS935 for their technical assistance and daily scientific exchanges. We are grateful to Ibrahim Casals and Benoit Peuteman from the INSERM UMS33 Animal Facility (SEIVIL). We appreciate the assistance of Marie-France Champy, Aurélie Auburtin, Tania Sorg and Yann Herault from Phenomin (Institut Clinique de la souris, 1 rue Laurent Fries, 67404 ILLKIRCH cedex 2; CNRS, UMR7104, Illkirch, France; INSERM, U964, Illkirch, France; Université de Strasbourg, France) with blood analyses. We thank Karine Bailly from the Cochin Cytometry and Immunobiology Facility (CYBIO, Institut Cochin, Paris) for performing the Luminex assay. We also thank Lindsay Higgins for her English editing services. We are grateful to Francis Levi and Angela Nieto for their helpful comments on the manuscript. This work was funded by Inserm, University Paris Sud, INRA, Association Institut de Cancérologie et d'Immunogénétique (ICIG), Vaincre le Cancer-NRB, Fond Avenir MASFIP, and GEFLUC – Les Entreprises contre le cancer. The post-doctoral fellowship of E.H. was granted by Vaincre le Cancer-NRB and the University Paris Saclay (Project BioTherAlliance).

## Author contributions

E.H., A.B.G. and H.A. designed the experiments. E.H. performed most of the in vitro and in vivo experiments with the support of W.T. Y.A. and S.P. performed the analysis of bone phenotypes. M.V., J.L., H.A. and I.R.L. performed the analysis of primary tumour histology. G.D. and C.A. were in charge of animal housing, breeding and manipulation. X.L. and S.D. provided all the protocols, equipment and support for chronobiology experiments. H.A. performed secondary analysis of mRNA-seq data. E.H., S.P., and H.A. wrote the paper. A.B.G. and H.A. supervised the research and funded the project.

## Competing interests

The authors declare no competing interests.
