## [Peer Review File · Nature Communications]

Reviewers' comments:

Reviewer #1 (Remarks to the Author): Expertise in breast cancer (in vivo) genomics and transcriptomics

In this manuscript, the authors hypothesize that disruption of circadian rhythm will impact tumor progression and metastasis. To test this, they have disrupted the light/dark cycle in PyMT transgenic mice. They claim that metastasis is altered, as is stemness, corresponding to a shift in the immune component of the tumor microenvironment. Use of a CXCR2 inhibitor was thought to "correct these defects".

While potentially interesting, there were a number of shortcomings in the manuscript that should be addressed. Several of these issues prevented validation of their conclusions.

1) The methods do not detail the background of the MMTV-PyMT mice, nor do they describe how the tumor studies were completed. The primary issue that, due to the lack of detailed methods, needs to be resolved is why the frequency of metastasis in LD PyMT mice is so low. Numerous previous studies have shown that virtually 100% of PyMT mice on the FVB background develop pulmonary metastases, not the 30(ish)% they show.

2) Color blind readers simply cannot see figures that are red-green. Figure 2C is an excellent example of a figure that does not need to be red-green, but because it is, it cannot be interpreted. It literally appears as the same shade for both top and bottom. Black and grey would work. Or simply a bar that shows % with metastasis. Other problem figures include:

Figure 3A

Figure 3B

Figure 3E

Figure 4F

Figure 6E

Also, supplemental data – please just change everything that is red/green.

3) Figure 2D should have dots representing the number of metastatic foci, not just bins of categories. Please redo statistical tests using all the data. In addition, representative images should be shown.

The statement that PC1 separates tumors with / without metastasis is unfounded based on the data in Figure 3B. In addition, there are far too few data points to even try this. This portion of the figure should be cut.

Figure 4A is not convincing for an increase of stem cells in the jet lag mice. P-values seem to be largely driven by a handful of outliers in the control population.

Figure 4F is based on 4 tumors from LD and 4 from JL – but NO characterization of the primary tumor was provided. There are numerous studies illustrating the heterogeneity in histology from PyMT mice. This was not taken into account and so the authors may be comparing spindleoid myoepithelial tumors to lobular epithelial tumors, with obvious bias in the percentage of potential tumor initiating cells.

CXCR2 inhibitors have previously been shown to inhibit metastasis. However, the experimental design in Figure 6 is lacking an essential control – no LD mice were tested +/- CDCX2 inhibitor to determine if the response for the JL and LD mice was altered. Indeed, given the work of Halpern et al (PMID 21601983), this control is essential. Also, it is essential to place the work into the context of this prior work.

Reviewer #2 (Remarks to the Author): Expertise on circadian rhythm and cancer

General comments

The biomedical applications of circadian clocks represent a critical challenge for medical progress, especially for cancer. More specifically, an increased risk of breast cancer has been shown in women undergoing prolonged shift work, and this environmental condition was acknowledged as a likely cause of cancer by the International Agency for Research on Cancer both in 2007 and in 2019. Several other reports show that circadian disruption also impacts on the outcomes of tumour bearing rodents, as well as large cohorts of cancer patients. Taken together, the existing literature emphasizes the need for a better understanding of the mechanisms linking circadian disruption and carcinogenesis. This manuscript provides highly interesting and innovative data in this regard.

In aggregate, Hadadi et al. highlight the impact of iterative daily schedule shifts ("chronic jet lag protocol") for the metastatic dissemination of breast cancer in an experimental model of spontaneous mammary carcinogenesis. They identify key mechanisms at work at the chemokine/cytokine network level, and within the tumour infiltrating immune cells. They show that a CXCR2 inhibitor could prevent the deleterious effect of iterative schedule shifts on breast cancer dissemination. The data presented are new and convincing, yet several issues deserve to be answered.

Main comments

- (1) As mentioned by the authors, the jet lag protocol that has been applied in their study has proven its ability to suppress circadian rhythms in rest-activity and core body temperature as well as several clock genes expressions, both in the SCN and in peripheral tissues of male B6D2F1 mice. Although it is highly likely that similar effects would be observed for the transgenic mice used in this study, it is needed to report the circadian phenotype of these mice when kept both in usual light-dark conditions (LD12:12), and on the "chronic jet lag" protocol.
- (2) Because the vast majority of the parameters studied undergo large endogenous circadian variations in mice on LD12:12 (as well as in constant darkness), it is essential that the sampling times and dosing times are reported in relation wto the Light-Dark schedule, as Zeitgeber Time.
- (3) Because the underlying assumption in the study is that the "chronic jet lag protocol" suppresses the circadian organisation, such timing reference is of a lesser importance if the assumption is proven in (1). However, it would be useful to know when the samples were taken in relation to the effective light or dark span the mice were exposed to.
- (4) Throughout the manuscript, there are some imprecisions, that are addressed in my specific comments
- (5) English should be improved.

Specific comments

Results

P3, last sentence refers to "slight tendency to observe more malignant lesions in JL mice". Table S1 and Fig S2 illustrate this sentence for 8 mice only (4 LD and 4 CJL). On which criteria were these 8 mice selected for pathology among the total of 46 that were on study?

P4, lines 8-10: a bone lesion is by definition abnormal. If the bone lesions were histologically proven as being metastatic deposits, better to say it as such.

P4, Line 11: rather say: "the proportion...increased from... to..."

P4, line 8 before the end: The down regulation of the light perception and phototransduction genes in the bone marrow mononuclear cells in "chronic jet lagged" mice needs to be interpreted against time-qualified expressions in controls. Is the expression of these genes known to be rhythmic? How large is their amplitudes?

P5, line 5: better say "statistically significant" rather than "slight" which implies a subjective interpretation of a non statistically significant difference, which is not the case according to Fig 4.

P5, line 24: when it comes to clock genes expression, it is crucial at least to report the ZT sampling time in the LD12:12 mice, and whether the sampling time in the "chronically jet lagged" mice also occurred at a similar time in the light or dark span as in the controls.

P5, second to last par.: You should speak of circadian clock disruption, but not of Per genes specifically, because there are no data in Per KO.

P6: sampling times are needed throughout...

P6, first line. Can you provide a brief statement as to why the reduced

P6, line 11: Can you discuss the evidence relating tumour CD4/CD8 to prognosis, and what it means in terms of the immunologic control of tumours. Isn't it enough to consider the CD8+ cells (suppressor/cytotoxic) that infiltrate the tumour?

P6, last sentences: How often was the CXCR2 inhibitor injected, through which route, at which ZT time/clock hour? Regarding timing, also see:

<https://www.ncbi.nlm.nih.gov/pmc/articles/PMC4967945/>

Figures

For several Figure panels, there is only a binary response, i.e. in Fig 2, panel C: Metastasis yes or no. It would be clearer to only show the rate of mice with metastases, and the SE of each percentage

First of all, we thank the two reviewers for their helpful comments and suggestions to improve the global quality of the manuscript.

Reviewer #1 (Remarks to the Author): Expertise in breast cancer (in vivo) genomics and transcriptomics

1) The methods do not detail the background of the MMTV-PyMT mice, nor do they describe how the tumor studies were completed. The primary issue that, due to the lack of detailed methods, needs to be resolved is why the frequency of metastasis in LD PyMT mice is so low. Numerous previous studies have shown that virtually 100% of PyMT mice on the FVB background develop pulmonary metastases, not the 30(ish)% they show.

Indeed, this important point was missing from the manuscript. We provided details and complete description in the main text (P3), methods (P10) and in Fig. 1A.

We did not use a pure PyMT FVB background because it was too aggressive and not compatible with our long-term chronic CRD protocol. We decided to use a mixed B6*FVB PyMT background. Using this background, we observed a delayed onset of tumour development and slower progression (Fig. 1A) with low prevalence (c.a. 30%) of lung metastasis at the age of 16 weeks.

2) Color blind readers simply cannot see figures that are red-green. Figure 2C is an excellent example of a figure that does not need to be red-green, but because it is, it cannot be interpreted. It literally appears as the same shade for both top and bottom. Black and grey would work. Or simply a bar that shows % with metastasis. Other problem figures include:

Figure	3A
Figure	3B
Figure	3E
Figure	4F
Figure	6E

Also, supplemental data – please just change everything that is red/green.

We thank the reviewer for drawing our attention to this point. We have changed the colours of the figures to make them interpretable for colour blind readers.

3) Figure 2D should have dots representing the number of metastatic foci, not just bins of categories. Please redo statistical tests using all the data. In addition, representative images should be shown.

We chose to represent our data in categories due to high variation. As you can see on the graphs below, two mice were detected with extremely high number of metastatic foci. Possible, these two mice reached the exponential late tumour growth phase. However, the tumour burden data suggest that only one of the mice reached the late phase. These samples were not excluded because:

- 1) From the same cohort/litter neither the 2 other JL nor the 3 LD mice developed lung metastasis**
- 2) Tumour burden did not correlate with number of foci (highlighted on the image below)**

3) None of the downstream analysis showed deviation from the median
Please note the Figure 2D and its statistic have been done using all data.
 Representative images can be found in Supplementary data (Supplementary Fig 4B)
 together with the distribution of metastatic foci number between groups, including the
 graph shown here.

The statement that PC1 separates tumors with / without metastasis is unfounded based on the data in Figure 3B. In addition, there are far too few data points to even try this. This portion of the figure should be cut.

We modified the figure and the text. From the previous figure 3B, PC1 separates primary tumours with/without metastasis only from JL mice (in red) but not from LD mice (black). We agree that this graph is a bit misleading and that the number of samples is reduced and cannot allow general conclusions to be drawn. We removed the PCA from the main figure and modified the text accordingly.

Figure 4A is not convincing for an increase of stem cells in the jet lag mice. P-values seem to be largely driven by a handful of outliers in the control population.

We would like to clarify that the stem population was identified based on the mouse mammary stem cell (MaSC) signature (Fig.4B) and not based on the individual expression of stemness markers. Importantly, we evaluated stemness by functional assays (mammosphere formation and tumour initiation). Both data confirmed our observation about enriched MaSC compartment in the JL tumours, which led us to the conclusion that CRD promotes stemness of primary tumour cells.

We agree that our data sets show relatively high variability but this is considered normal in *in vivo* data sets. Important to note that

- standard deviation (SD) /variability is not consistently different between LD vs JL group and LD SD is not consistently higher compared to JL SD
- Data sets were tested to identify outliers (ROUT method Q=1%, GraphPad):
CD24% data set: one outlier in LD group
CD29% data set: one outlier in JL group
No other outliers were detected which also confirms that the observed variability is not due to a specific group of mice.
Elimination of the two outliers do not alter the statistical analysis results, therefore we decided to keep both outliers.

Figure 4F is based on 4 tumors from LD and 4 from JL – but NO characterization of the primary tumor was provided. There are numerous studies illustrating the heterogeneity in histology from PyMT mice. This was not taken into account and so the authors may be comparing spindleoid myoepithelial tumors to lobular epithelial tumors, with obvious bias in the percentage of potential tumor initiating cells.

We agree with the reviewer that this is an important point. The histology of the primary tumours used for the tumour initiation study (6 primary tumours from LD and 6 primary tumours from JL mice) were not included in the first version of the manuscript. We did not see striking anatomical differences when we dissected the tumours to be used for the tumour initiation study but to confirm this empirical observation we also performed and included HES staining of the tumours we used in the revised manuscript. The tumours have been analysed by a veterinary pathologist (Isabelle Raymond Letron, professor in histology and pathology at the University of Toulouse and a recognized expert in veterinary pathology). The information is shown on Supplementary Table 2. Unsurprisingly, JL tumours are slightly more aggressive (Supplementary Fig. 3) but JL and LD tumours were quite homogenous and all arise from epithelial cells. We did not observed spindleoid myoepithelial tumours in these samples.

CXCR2 inhibitors have previously been shown to inhibit metastasis. However, the experimental design in Figure 6 is lacking an essential control – no LD mice were tested +/- CDCX2 inhibitor to determine if the response for the JL and LD mice was altered. Indeed, given the work of Halpern et al (PMID 21601983), this control is essential. Also, it is essential to place the work into the context of this prior work.

We agree that we have to put our results into the context of the mentioned Halpern et al 2011 and also the recently published Romero-Moreno et al 2019 (doi: 10.1038/s41467-019-12108-6) studies. The increased homing of DCCs to the bone and enhanced Cxcl5 gene expression in bone marrow (data was not shown in the first

version, new Supplementary Figure 9D) support the role and importance of CXCR2 in bone colonisation. Indeed, the treatment of CXCR2 inhibitor definitely could affect this aspect of metastatic spread. We addressed this on P7, P10 and Supplementary Fig. 9D.

However, it is important to note that our flow cytometry analysis showed no difference in the proportion of CXCR2+ primary tumour cells between LD and JL mice (Supplementary Fig. 9C). In addition, data from Luminex assay did not show altered LIX/CXCL5 level in plasma from JL mice compared to the control group. All together these data suggest that in our model CXCR2/CXCL5 axis does not drive directly the invasion and intravasation of tumour cells from the primary tumour site. Based on these observations we expected that CXCR2 inhibition primarily decreases metastasis through blocking myeloid cell recruitment and consequently improving anti-tumour immunity / reducing the development of pre-metastatic niche as it have been shown before (Steele et al 2016 DOI: 10.1016/j.ccell.2016.04.014, Acharyya et al 2011 DOI 10.1016/j.cell.2012.04.042) in standard/LD conditions. Altogether, in line with the 3R paradigm, originally we did not include the LD group in this 'proof-of-concept' study because: (1) the CXCR2 inhibition effect on metastasis is well described and (2) we observed our JL mice responding similar way as it has been described in previous studies.

Indeed, the results of the requested experiment show similar changes upon CXCR2 inhibition in LD condition. We observed reduced metastatic dissemination and a shift in tumour immune microenvironment with reduced myeloid cells and CD4/CD8 ratio in CXCR2 inhibitor treated mice (Supplementary Fig. 10) Due to the slow tumour progression to late phase in our tumour model under LD condition (at least 20 weeks) we used F1 hybrid generation of PyMT-FVB * Luc2-B6 mice. These mice show slower tumour progression compared to PyMT-FVB but significantly faster compared to the mice used for our JL model.

We agree that the use of CXCR2 inhibition is not specifically counteracting the effects of JL but we propose it as a possible complementary therapy to control the speed-up of tumour progression in CRD conditions. This point was included in the discussion (P10).

Reviewer #2 (Remarks to the Author): Expertise on circadian rhythm and cancer

General comments

The biomedical applications of circadian clocks represent a critical challenge for medical progress, especially for cancer. More specifically, an increased risk of breast cancer has been shown in women undergoing prolonged shift work, and this environmental condition was acknowledged as a likely cause of cancer by the International Agency for Research on Cancer both in 2007 and in 2019. Several other reports show that circadian disruption also impacts on the outcomes of tumour bearing rodents, as well as large cohorts of cancer patients. Taken together, the existing literature emphasizes the need for a better understanding of the mechanisms linking circadian disruption and carcinogenesis. This manuscript provides highly interesting and innovative data in this regard.

In aggregate, Hadadi et al. highlight the impact of iterative daily schedule shifts ("chronic jet lag protocol") for the metastatic dissemination of breast cancer in an experimental model of

spontaneous mammary carcinogenesis. They identify key mechanisms at work at the chemokine/cytokine network level, and within the tumour infiltrating immune cells. They show that a CXCR2 inhibitor could prevent the deleterious effect of iterative schedule shifts on breast cancer dissemination. The data presented are new and convincing, yet several issues deserve to be answered.

Main comments

(1) As mentioned by the authors, the jet lag protocol that has been applied in their study has proven its ability to suppress circadian rhythms in rest-activity and core body temperature as well as several clock genes expressions, both in the SCN and in peripheral tissues of male B6D2F1 mice. Although it is highly likely that similar effects would be observed for the transgenic mice used in this study, it is needed to report the circadian phenotype of these mice when kept both in usual light-dark conditions (LD12:12), and on the “chronic jet lag” protocol.

We performed locomotor activity and core body temperature measurement. Our data confirmed the circadian disruption effect of the applied jetlag protocol in our transgenic model. Results are added to the main figure, text (P4) and supplementary data (Fig. 1, Supplementary Figure 1 and Supplementary Table 1).

(2) Because the vast majority of the parameters studied undergo large endogenous circadian variations in mice on LD12:12 (as well as in constant darkness), it is essential that the sampling times and dosing times are reported in relation to the Light-Dark schedule, as Zeitgeber Time.

(3) Because the underlying assumption in the study is that the “chronic jet lag protocol” suppresses the circadian organisation, such timing reference is of a lesser importance if the assumption is proven in (1). However, it would be useful to know when the samples were taken in relation to the effective light or dark span the mice were exposed to.

We agree with the reviewer and as mentioned before, we validated our “chronic jet lag protocol “ using locomotor activity and core body temperature measurement. Our data confirmed the circadian disruption effect of the applied jetlag protocol in our transgenic model. For LD mice, and in response to (2) and (3): we clarified this in the methods and added the respective ZT times.

(4) Throughout the manuscript, there are some imprecisions, that are addressed in my specific comments

The imprecisions have been corrected. See our answers to specific comments

(5) English should be improved.

The manuscript was reread by a professional scientific writer.

Specific comments

Results

P3, last sentence refers to “slight tendency to observe more malignant lesions in JL mice”. Table S1 and Fig S2 illustrate this sentence for 8 mice only (4 LD and 4 CJL). On which criteria were these 8 mice selected for pathology among the total of 46 that were on study?

We selected the mice from random selection based on sample availability. Histology was performed in the middle of cohort collection, 8 samples were picked from 20 mice. We agree that this number was low and we analysed the histology of tumours from 9 additional mice and completed the previous table S1 (now Supplementary Table 2). The new dataset confirms and strengthens the increased aggressiveness of tumours from JL mice (see new graph on Supplementary Fig. 3)

P4, lines 8-10: a bone lesion is by definition abnormal. If the bone lesions were histologically proven as being metastatic deposits, better to say it as such.

Addressed and we modified the text accordingly.

P4, Line 11: rather say: "the proportion...increased from... to..."

We corrected the text.

P4, line 8 before the end: The down regulation of the light perception and phototransduction genes in the bone marrow mononuclear cells in "chronic jet lagged" mice needs to be interpreted against time-qualified expressions in controls. Is the expression of these genes known to be rhythmic? How large is their amplitudes?

This is an important point and we were really surprised to observe down regulation of phototransduction genes in internal tissues and cells. Indeed, we can not exclude that all these genes harbour the same circadian rhythmic expression peaking around ZT3-ZT4, when LD mice were sampled. In this case, it is logical that these genes appeared downregulated in JL mice (in relation with the amplitude of their rhythmic expression), where the core circadian clock is disrupted. This important point has to be confirmed, as little is known about the expression/function of these genes in peripheral non-visual tissues. In our mind it would be strange that all these genes express the same rhythmicity (if they are rhythmic) but we modified the discussion in relation to this point. Moreover another important question relies on the biological consequences of such downregulation.

P5, line 5: better say "statistically significant" rather than "slight" which implies a subjective interpretation of a non statistically significant difference, which is not the case according to Fig 4. **We corrected the text.**

P5, line 24: when it comes to clock genes expression, it is crucial at least to report the ZT sampling time in the LD12:12 mice, and whether the sampling time in the "chronically jet lagged" mice also occurred at a similar time in the light or dark span as in the controls.

We corrected the text.

P5, second to last par.: You should speak of circadian clock disruption, but not of Per genes specifically, because there are no data in Per KO.

Addressed and we modified the text accordingly.

P6: sampling times are needed throughout....

We added ZT, collection times.

P6, first line. Can you provide a brief statement as to why the reduced

Most probably, in our experimental system the reduced number of TICs were due to the disrupted diurnal trafficking of leukocytes. As Zhao et al 2017 showed under circadian disruption leukocytes lost their rhythmic trafficking leading to a decrease in the daily total number of circulating leukocytes. Other mechanisms linked to increased tumour burden (e.g. hypoxia or necrosis) could also contribute to this reduction. We added a sentence in the discussion (P9).

P6, line 11: Can you discuss the evidence relating tumour CD4/CD8 to prognosis, and what it means in terms of the immunologic control of tumours. Isn't it enough to consider the CD8+ cells (suppressor/cytotoxic) that infiltrate the tumour? **Number of TILs and the ratio between different TIL subpopulations have been long time investigated in relation to tumour immunity monitoring and therapy responsiveness/prognosis (Gisterek et al 2009 10.1016/S1507-1367(10)60011-9, Gooden et al doi:10.1038/bjc.2011.189). CD4/CD8 ratio showed association with relapse free survival and overall survival (Wang et al 2017 10.1016/j.humpath.2017.09.012) Higher ratio of CD8/CD4 counts were associated with pathologic complete response (pCR) (Castaneda CA et al 10.5306/wjco.v7.i5.387,). Also Garcia-Martinez et al showed that after neoadjuvant chemotherapy the inversion of CD4/CD8 ratio (CD4+ TIL decreased, CD8+ TIL increased) was associated with pCR and better prognosis (10.1186/s13058-014-0488-5).** **Number of TILs can be highly variable between individuals therefore additional use of CD4/CD8 ratio can provide cleaner results and further support the CD8+ TIL data (this is more obvious on the peripheral blood data Fig. 5F). We added a sentence in the results' section to clarify this point (P6).**

P6, last sentences: How often was the CXCR2 inhibitor injected, through which route, at which ZT time/clock hour? Regarding timing, also see: <https://www.ncbi.nlm.nih.gov/pmc/articles/PMC4967945/>

The injection protocol is detailed in method section: through 8 weeks, on each week intra peritoneal injection was performed once daily 5 days in a row followed by 2 days resting (adapted from Acharyya et al 2012 DOI: 10.1016/j.cell.2012.04.042) LD mice were injected at ZT6-ZT7, at the previously described peak of circulating CD45+ cells (Zhao et al 2017 DOI 10.1182/blood-2017-04-778779). Regarding the timing in JL mice: based on the same study from Zhao et al 2017, which showed no significant circadian oscillation of leukocyte trafficking in mice exposed to chronic jetlag, we did not apply specific ZT time for injection. However, mice were consistently injected at the same time of the day to keep daily dosing. We clarified this in the method section.

Figures

For several Figure panels, there is only a binary response, i.e. in Fig 2, panel C: Metastasis yes or no. It would be clearer to only show the rate of mice with metastases, and the SE of each percentage

To address this comment we altered the visual appearance of all the plot graphs (eg. Fig 2C and 2D) but we kept the original analysis method.

Reviewers' comments:

Reviewer #1 (Remarks to the Author):

Revised manuscript addresses all points satisfactorily.

Reviewer #2 (Remarks to the Author):

My comments and questions have been addressed properly well.

A few specific comments:

- Legend to Fig 1 (A): Schematic design of tumour progression study in FVB.... Rather than Schematic graph of tumour progression in FVB.... The representative actograms should appear as a separate panel (B); other panel ID's should be shifted accordingly
- Legend to Figure 2: It might be helpful to recall the sampling were done at 16 weeks of age
- Legend to Fig 4 (D): Circadian clock genes mRNA expressions (if this was the case)
- Legend to Suppl Fig 1: Please use real study days (age) rather than the arbitrary ones (starting recording). Any reason why not to show the temperature profiles?

Reviewer #3 (Remarks to the Author):

This reviewer was asked to comment on the immunological aspects of the work. However, given that this work falls on my area of expertise, I also made comments on the rest of the manuscript.

Figure 1

Major:

Photon flux is acceptable to show tumor initiation, but it does not accurately reflect tumor growth: Luciferase is driven by MMTV in a constant manner, independently of whether those cells become transformed and contribute to tumor formation. In other words, many of those Luciferase-positive cells will not form tumors. Moreover, the luciferase signal becomes rapidly saturated in this model, and does not allow accurate quantification of tumor growth. Tumor volume should be used instead.

Minor:

It is unclear why in the last panel (1F) the value is a % between tumor weight and mouse weight. It is customary to plot combined tumor weight per mouse when using transgenic mice.

The way Figure 1 is presented is confusing, and it gives the wrong impression that the jet-lag model was applied to the mixed-background, while the FvB mice were kept under regular dark-light conditions. Please clarify.

The tumor growth kinetic difference between the two backgrounds is known and established, and the reference is enough to justify their election. It is my suggestion to eliminate this panel, to avoid confusion on the utilized models.

It is not clear what "mixed background" refers to. If this is the F1 between the FvB and B6 mice, in which there is 50% of each background, this would be acceptable, but not if a random number of crosses was done between the 2 strains.

Figure 2

Major:

The Santa Cruz PyMT antibody is notoriously poor, and figure 2A-B indicate it was used to gate on the PyMT positive cells in the bone marrow by flow. Representative images of the flow plot are necessary to show how reliable this data is.

Minor:

It is not clear what the point of showing bone marrow mets is. This data is not quantified, and therefore it does not contribute to the differential effect on metastasis the authors are trying to establish. Consider removing.

Supplementary Tables 3 and 4 are missing.

Figure 3

Major

Very weak figure. There is no significant conclusion drawn, nor use of this data. Maybe supplementary? As is, it only disrupts the flow of the manuscript and does not contribute in any meaningful way.

Figure 4

Major

Figure 4A does not provide any evidence for enrichment of stem cells, as none of these markers means anything in isolation. It should be removed, or converted into the appropriate marker combinations to complete figure 2B with additional definitions for mammary stem cells.

Figure 4E: Isn't the role of Per2 in stemness already described? If so, this experiment is simply confirmatory.

Figure 4F poses a problem: if JL treatment increases stemness, tumor initiation and growth, then it is not clear why the primary tumors in the PyMT mice are not significantly affected, but only metastasis.

Figure 5-Immunology comments

Major

In identifying the references for their flow cytometrical analysis of tumor-infiltrating leukocytes, the authors correctly point to two landmark papers pioneering this classification in murine breast tumors, and more specifically PyMT tumors: Movahedi et al, Cancer Research, 2010; and Franklin et al, Science, 2014. However, their gating strategy does not follow any of the ones described in the references. While there are many different ways to analyze the TIL populations, the main issue is the order of gating, using less cell type-specific markers first, and eliminating important populations from subsequent analysis with more accepted markers. For example, the first gating after identifying hematopoietic cells (CD45+) should be CD11B and CD3/TCRB to identify myeloid cells and T cells, respectively, and then continue gating on each individual population.

A non-exhaustive list of other issues:

*While the CD45+ CD11B+ Gr1+ cell population contains neutrophils, not all cells here are neutrophils, and therefore this is not how they should be named. In fact, this is how the community defines immature myeloid cells, also called myeloid-derived suppressor cells (MDSCs).

*The sub classification of TAMs based on MHCII levels is usually done after discrimination of Ly6C (monocytic) cells, as properly depicted in Movahedi et al.

*Monocytes are defined as CD45+ CD11B+ Ly6C+ cells. There is no Ly6C staining done in this analysis, and therefore monocytes cannot be properly identified.

*NK cells are usually defined as NK1.1+ cells, and separated in NK or NKT based on their

expression of the T cell receptor. Here, NK cells are identified as CD45+ CD11B+Gr1- CD11C+MHCII-CD64+ cells...none of these markers are NK-specific, and seem more of a random collection.

*B cells are named, but there are no B cell specific markers (like CD20 or B220)
Etc...

While it is still possible that there are changes consistent with a more pro-tumorigenic immune microenvironment, the cell populations need to be better defined.

Figure 5E-F

It is well established that increased CD8T cell infiltration is a marker of better prognosis in breast (and other cancers). It is also well established that CD4+ Foxp3+ Treg cells are a marker for poor prognosis in breast (and other cancers). The correlation of CD4T cells without any additional marker is, however, very controversial. This is simply because CD4T cells are heterogeneous: CD4+ Foxp3+ Treg cells are a marker of bad prognosis, TH2 CD4T cells induce alternative activation of macrophages, which is associated with poor prognosis, and TH1 CD4T cells are anti-tumorigenic, to name a few. Therefore, Treg cell frequencies and their relative ratio with respect to CD8T cells, would be a better correlate for outcome than generic CD4/CD8 ratios.

Supplementary Figure 5

With exception of IL-4, none of the other changes are significant. The comment of the elevation of CXCL12 levels in the JL mice should be removed from the text, because equally irrelevant observations in the opposite direction can be made from that table (For example increase in IL-12, which is an inducer of TH1 responses, antitumorigenic phenotypes). Furthermore, IL-4 is a critical cytokine inducing alternative activation of tumor-associated macrophages, which argues the JL tumors have a tumor-promoting immune phenotype. Unfortunately, this data not only does not help the main conclusion the authors are trying to make, but also argues against it. The authors would have been better off discarding this due to the high biological variability observed.

Figure 6

Major

Expression data shown in Figure 6A (from RNASeq experiment) does not indicate which changes are statistically significant, nor what cut-off value was used for the analysis. Several of these transcripts do not seem to be significantly changed, like CXCL11 and CXCL9. Interestingly, those are bona fide and highly sensitive targets of IFN γ , which the authors claim is significantly downregulated (p values?).

Other transcripts with opposing functions look like they could possibly be significantly upregulated in both models. For example, IL1s (alpha and beta – pro-tumorigenic factors) and IL-10 (immune-suppressive factor) seem upregulated in JD mice.

All of these data seem to be “selectively” interpreted to fit the proposed hypothesis. Perhaps a pathway analysis could provide more unbiased support to their hypothesis?

IN SUMMARY: the rationale for selecting CXCR2 for further studies is convoluted and lacks rigor. The authors should make the effort to much better justify the target.

Major

The fact that the CXCR2 inhibition works to the same degree in jet-lagged mice as in non jet-lagged mice suggests that this pathway is not selective to circadian regulation of metastatic behavior in these mice. If it is not, then we are still lacking an explanation for the main observation of the paper. The discussion is not sufficient to explain this fact.

We thank all the reviewers for their help in improving the quality of this research. In this last version, we would like to thank the last reviewer for his constructive comments and suggestions, particularly regarding the immunological aspect of this work.

Reviewer #1 (Remarks to the Author):

Revised manuscript addresses all points satisfactorily.

Reviewer #2 (Remarks to the Author):

My comments and questions have been addressed properly well.

A few specific comments:

- Legend to Fig 1 (A): Schematic design of tumour progression study in FVB.... Rather than Schematic graph of tumour progression in FVB....

The representative actograms should appear as a separate panel (B); other panel ID's should be shifted accordingly

- Legend to Figure 2: It might be helpful to recall the sampling were done at 16 weeks of age

- Legend to Fig 4 (D): Circadian clock genes mRNA expressions (if this was the case)

- Legend to Suppl Fig 1: Please use real study days (age) rather than the arbitrary ones (starting recording). Any reason why not to show the temperature profiles?

We addressed these specific comments. We included the temperature profiles in the Supplementary Figure 1 and text. The respective age of mice is now presented in methods and in figure and table legends.

Reviewer #3 (Remarks to the Author):

This reviewer was asked to comment on the immunological aspects of the work. However, given that this work falls on my area of expertise, I also made comments on the rest of the manuscript.

Figure 1

Major:

Photon flux is acceptable to show tumor initiation, but it does not accurately reflect tumor growth: Luciferase is driven by MMTV in a constant manner, independently of whether those cells become transformed and contribute to tumor formation. In other words, many of those Luciferase-positive cells will not form tumors. Moreover, the luciferase signal becomes rapidly

saturated in this model, and does not allow accurate quantification of tumor growth. Tumor volume should be used instead.

We do not have the data on tumour volume for these mice. We clearly stated in the main text that we used in vivo bioluminescence to monitor tumor growth (p.4). We agree that luciferase measurement does not allow accurate quantification of tumour growth and that it is possible that cells expressing luciferase are not all actively proliferating and contributing to tumour development.

However, we also observed gradual increase of signal with tumour growth supporting the idea that the majority of signals are coming from proliferating cancer cells (see the dedicated Figure below showing luciferase imaging in the same mice at different times of tumor development). Moreover, in our experimental design, we did not observe saturation of the luciferase signal mostly because we focused on early phases of tumor development. In addition measuring luciferase in vivo also allowed us to monitor tumor growth even when, as was sometimes the case, palpable tumors appeared late (14 weeks) and when we observed non-homogenous tumour growth in-between mammary fat pads.

Furthermore, the information shown on Figure 1E-F is mostly confirmative since previous studies have already shown the adverse effect of CRD/dysfunctional circadian rhythm on tumour initiation and growth in different cancer models (Papagiannakopoulos et al 2016 doi: 10.1016/j.cmet.2016.07.001 , Van Dycke et al 2015 doi: 10.1016/j.cub.2015.06.012 , Kiessling et al 2017 doi: 10.1186/s12915-017-0349-7 , Filipski et al 2004 doi: 10.1158/0008-5472.CAN-04-0674).

Figure: Follow up by in vivo imaging of tumour growth in 2 mice from 10 weeks to 16 weeks.

Minor:

It is unclear why in the last panel (1F) the value is a % between tumor weight and mouse weight. It is customary to plot combined tumor weight per mouse when using transgenic mice.

We decided to represent total tumour weight as tumour burden to remove the potential biased effect of mouse weight. CRD was shown to result in increased weight gain/obesity (Van Dycke et al 2015 , Shi et al 2013 10.1016/j.cub.2013.01.048 , Thaïss et al 2014 10.1016/j.cell.2014.09.048) and obesity has been linked to enhanced tumour growth (Lengyel et al 10.1016/j.trecan.2018.03.004) However, we did not observe significant increase in body weight in our JL cohort (Fig 1C).

Like Fig.1G, the figure below without body weight normalisation illustrates a significant increase of total tumour weight in JL mice compared to LD mice (unpaired t-test):

The way Figure 1 is presented is confusing, and it gives the wrong impression that the jet-lag model was applied to the mixed-background, while the FvB mice were kept under regular dark-light conditions. Please clarify.

We totally agree and we simplified the graph on Figure 1A to avoid confusion.

The tumor growth kinetic difference between the two backgrounds is known and established, and the reference is enough to justify their election. It is my suggestion to eliminate this panel, to avoid confusion on the utilized models.

We simplified the graph on Figure 1A to avoid confusion.

It is not clear what "mixed background" refers to. If this is the F1 between the FvB and B6 mice, in which there is 50% of each background, this would be acceptable, but not if a random number of crosses was done between the 2 strains.

The first reviewer previously raised this point and we provided details in Method section p.11.

Figure 2

Major:

The Santa Cruz PyMT antibody is notoriously poor, and figure 2A-B indicates it was used to gate on the PyMT positive cells in the bone marrow by flow. Representative images of the flow plot are necessary to show how reliable this data is.

The Figure 2A and 2B have been modified and show the representative flow plots.

Also please find below the same PBMC plots as dot plots, to have a clear view on actual sample sizes with representative gating.

Regarding the antibody, we used the sc-53481 PyMT antibody for our whole lung stainings and we observed specific stainings from the beginning. As we had no issue with this antibody therefore we used its conjugated form for flow cytometry analysis. This antibody used in several peer reviewed publications including high impact journals (Nature Cell Biology doi: 10.1038/ncb3434 or EMBO Molecular Medicine doi: 10.1002/emmm.201201546).

Minor:

It is not clear what the point of showing bone marrow mets is. This data is not quantified, and therefore it does not contribute to the differential effect on metastasis the authors are trying to establish. Consider removing.

As the PyMT model is not a classical bone metastasis model we aimed to further confirm our flow cytometry and qPCR data on tumour cell homing and colonisation to the bone.

Supplementary Tables 3 and 4 are missing.

Supplementary Tables 3 and 4 are not in the supplementary information but are provided as supplementary data.

Figure 3

Major

Very weak figure. There is no significant conclusion drawn, nor use of this data. Maybe supplementary? As is, it only disrupts the flow of the manuscript and does not contribute in any meaningful way.

We agree that the mRNA-seq study did not bring clear global information, due to the low number of DEGs between conditions. We simplified the figure and focused on phototransduction genes because this point seems interesting to us.

Figure 4

Major

Figure 4A does not provide any evidence for enrichment of stem cells, as none of these markers means anything in isolation. It should be removed, or converted into the appropriate marker combinations to complete figure 2B with additional definitions for mammary stem cells.

We agree with the reviewer and changed this figure: We placed Figure 4A in supplementary Figure 6A and instead we added a representative gating strategy of MaSC population to complete figure 4B as suggested.

Figure 4E: Isn't the role of Per2 in stemness already described? If so, this experiment is simply confirmatory.

Previous studies linked circadian clock genes' function to stemness. Indeed, downregulation of Per2 in MCF10A breast cell line increased cells' stemness. However, this experiment does not make it possible to discriminate whether the phenotype results from an alteration of the circadian clock linked to the downregulation of Per2 or to a direct effect of Per2 on the stemness of breast epithelial cells. Our results showed that the phases of the functional circadian clock are intrinsically able to modulate the stemness of human mammary epithelial cells (Figure 4E). These two aspects are complementary and consistent: we observed here that human breast cells present more stemness during the

“night” phase when Per2 is low and Bmal1 is high and their stemness decreases during the “day” phase, when Per2 is high and Bmal1 is low.

Figure 4F poses a problem: if JL treatment increases stemness, tumor initiation and growth, then it is not clear why the primary tumors in the PyMT mice are not significantly affected, but only metastasis.

Indeed, we observed more drastic changes in metastatic spread but primary tumours were also affected. Primary tumours from JL mice were significantly bigger (Figure 1G, see also the total tumour weight graph provided above) and they were classified with significantly higher tumour grades (Supplementary Fig.3 and Supplementary Table 2) which correlates with the increased tumour initiation capacity of JL tumour cells.

Figure 5-Immunology comments

Major

In identifying the references for their flow cytometrical analysis of tumor-infiltrating leukocytes, the authors correctly point to two landmark papers pioneering this classification in murine breast tumors, and more specifically PyMT tumors: Movahedi et al, Cancer Research, 2010; and Franklin et al, Science, 2014. However, their gating strategy does not follow any of the ones described in the references. While there are many different ways to analyze the TIL populations, the main issue is the order of gating, using less cell type-specific markers first, and eliminating important populations from subsequent analysis with more accepted markers. For example, the first gating after identifying hematopoietic cells (CD45+) should be CD11B and CD3/TCRB to identify myeloid cells and T cells, respectively, and then continue gating on each individual population.

A non-exhaustive list of other issues:

*While the CD45+ CD11B+ Gr1+ cell population contains neutrophils, not all cells here are neutrophils, and therefore this is not how they should be named. In fact, this is how the community defines immature myeloid cells, also called myeloid-derived suppressor cells (MDSCs).

*The sub classification of TAMs based on MHCII levels is usually done after discrimination of Ly6C (monocytic) cells, as properly depicted in Movahedi et al.

*Monocytes are defined as CD45+ CD11B+ Ly6C+ cells. There is no Ly6C staining done in this analysis, and therefore monocytes cannot be properly identified.

*NK cells are usually defined as NK1.1+ cells, and separated in NK or NKT based on their expression of the T cell receptor. Here, NK cells are identified as CD45+ CD11B+Gr1- CD11C+MHCII-CD64+cells...none of these markers are Nk-specific, and seem more of a random collection.

*B cells are named, but there are no B cell specific markers (like CD20 or B220)

Etc...

While it is still possible that there are changes consistent with a more pro-tumorigenic immune microenvironment, the cell populations need to be better defined.

Thank you, indeed this section was missing essential information.

First, we would like to clarify that we made a significant mistake as we missed out to cite the reference paper of our gating strategy in our previous submitted manuscript version. We corrected this. Our primary gating strategy of tumour infiltrating leukocytes is based on the publication of Yu et al. 2016 (doi: 10.1371/journal.pone.0150606). We chose this method as it provides one relatively simple panel to identify the main immune cell types, which was adjustable to our 8-colour flow cytometer and allowed to perform complete characterisation even in case of limited sample size. We agree the used method is less accurate for lymphoid populations and it is a rather unorthodox approach. However, it showed to work well to identify different myeloid cell populations in non-lymphoid tissues including PyMT mammary tumours (Ye et al. 2016). As based on this gating strategy we only concluded information for tumour associated macrophages and we are convinced that the strategy to define TAM populations is correct.

Indeed, we modified the main text and figures to be more precise on gate naming.

In our staining we used REA526 clone (Miltenyi) of Ly6G/Gr-1 antibody, which clone is completely overlapping with 1A8 and known to only react with Ly6G. MDSCs can be divided into PMN-MDSCs (CD11b+Ly6G+Ly6Cl_o) and M-MDSCs (CD11b+Ly6G-Ly6Chi₊). Based on this we renamed our neutrophil gate to Neutrophils/PMN-MDSCs and we also changed Gr-1 to Ly6G to avoid confusion.

In our gating, instead of Ly6C we used MHCII expression vs side scatter to exclude monocytes. The CD11b+CD11c₊ cells (G6) were divided into MHC II₋/SSC_{low} population (G9), which contains monocytes, NK cells and also CD11c₊ T lymphocytes, and into MHCII₊/SSC_{int/hi} macrophages, DCs and eosinophils. We renamed Monocytes to Monocytic cells as it may include M-MDSCs. However, in tumours M-MDSCs showed rapid differentiation to TAM (Gabrilovich 2017 10.1158/2326-6066.CIR-16-0297) suggesting the majority of these cells can be found in our TAM gate.

Regarding NK, T, B cells we did not use specific markers for main immune cell characterisation based on Yu et al gating strategy where the gated cells were confirmed by specific markers.

To identify T cell phenotypes we used a separate antibody panel based on classical markers. Gating has been added to Supplementary Figure 7B.

We also provided a detailed antibody list (supplementary Table 7).

Figure 5E-F

It is well established that increased CD8T cell infiltration is a marker of better prognosis in breast (and other cancers). It is also well established that CD4+ Foxp3+ Treg cells are a marker for poor prognosis in breast (and other cancers). The correlation of CD4T cells without any additional marker is, however, very controversial. This is simply because CD4T cells are heterogenous: CD4+ Foxp3+ Treg cells are a marker of bad prognosis, TH2 CD4T cells induce alternative activation of macrophages, which is associated with poor prognosis, and TH1 CD4T cells are anti-tumorigenic, to name a few. Therefore, Treg cell frequencies and their relative ratio with respect to CD8T cells, would be a better correlate for outcome than generic CD4/CD8 ratios.

Indeed, we agree that it is more informative to use CD4+FoxP3+ Treg to CD8T cell ratio and it has a stronger prognostic value. We did perform FoxP3 immunostaining on primary tumours and we added the panel to the revised version (Figure 5F). Due to sample size limitation this staining was not performed on PBMCs and on tumours from CXCR2 inhibitor study. Gating strategy is presented in Supplementary Figure 7B.

Supplementary Figure 5

With exception of IL-4, none of the other changes are significant. The comment of the elevation of CXCL12 levels in the JL mice should be removed from the text, because equally irrelevant observations in the opposite direction can be made from that table (For example increase in IL-12, which is an inducer of TH1 responses, antitumorigenic phenotypes). Furthermore, IL-4 is a critical cytokine inducing alternative activation of tumor-associated macrophages, which argues the JL tumors have a tumor-promoting immune phenotype. Unfortunately, this data not only does not help the main conclusion the authors are trying to make, but also argues against it. The authors would have been better off discarding this due to the high biological variability observed.

We agree that globally, these results were disappointing and are only shown on a supplementary figure. Following the reviewer's suggestion we removed the comment about CXCL12 level as indeed it did not provide additional information to our results.

Regarding IL-4 data, we are aware that it is a critical cytokine to induce TH2 type immune responses but we would like to highlight several point:

- 1) Plasma cytokine/chemokine levels are not necessarily representing the tumour microenvironment.**
- 2) Plasma level of several cytokines/chemokine including IL-4 shows circadian fluctuation (most recent publication: 10.1038/s41598-019-56951-5). Cortisol, which is able to increase IL-4 plasma level is also under circadian regulation. Altogether, these suggest that the observed decrease can be due to circadian disruption.**

- 3) IL-4 plasma level showed only a moderate decrease in JL mice (0.8 fold), which might be well within the range of its circadian fluctuation. It is unreasonable to think that this level of decrease would completely impaired TH2 type immune responses.
- 4) Increase of pro-tumour immune phenotypes in JL tumours support the idea of CRD-induced shift towards pro-tumour microenvironment.

Figure 6

Major

Expression data shown in Figure 6A (from RNA-Seq experiment) does not indicate which changes are statistically significant, nor what cut-off value was used for the analysis. Several of these transcripts do not seem to be significantly changed, like CXCL11 and CXCL9. Interestingly, those are bona fide and highly sensitive targets of IFN γ , which the authors claim is significantly downregulated (p values?).

Other transcripts with opposing functions look like they could possibly be significantly upregulated in both models. For example, IL1s (alpha and beta – pro-tumorigenic factors) and IL-10 (immune-suppressive factor) seem upregulated in JD mice.

All of these data seem to be “selectively” interpreted to fit the proposed hypothesis. Perhaps a pathway analysis could provide more unbiased support to their hypothesis?

IN SUMMARY: the rationale for selecting CXCR2 for further studies is convoluted and lacks rigor. The authors should make the effort to much better justify the target.

We addressed this comment by rewriting the cytokine-chemokine network result section (page 7). We also modified Figure 6A by adding cut-off values to the boxplot and we added a supplementary table (supplementary Table 6) reporting FPKM expression values, log₂FC and p-values for all the genes that appeared on boxplots from Figure 4D, Figure 6A and Supplementary Figure 9, including selected chemokines/cytokines. We also included in this table gene expression mean and p-values obtained from DESeq2, the software used for differential gene expression analysis. For Figure 6A, FPKM based Log₂FC values were used to select the most down-(cut-off: <-0.8) and upregulated (>0.8) cytokines/chemokines. For data interpretation both FPKM and DESeq2 based results with respective p-values were taken into consideration. IFN γ showed significant downregulation in both analysis method (DESeq2 -1.05 Log₂FC, p=0.044, FPKM -1.58 Log₂FC, p=0.029) while its targets Cxcl9 and Cxcl11 had <-0.8Log₂FC change in their expression but it did not pass statistical significance. Indeed, both IL1s and IL10 showed significant upregulation (Supp. Table 6).

We provide below a more logical and cohesive reasoning for the rationale behind our study flow and target selection.

In summary, CXCL5, which happened to be upregulated in both primary tumours and BM of JL mice (Figure 6A and Supplementary Figure 9B and 9E), and its receptor CXCR2 have

been previously described to be under circadian regulation (Gibbs J et al 2014 Nat. Med., He W et al 2018 Immunity). The enrichment of CXCR2+ immune cells in primary tumours support the idea of enhanced CXCR2 axis in JL mice. Furthermore, the role of CXCR2 axis in the recruitment of MDSCs (Gabrilovich DI & Nagaraj S 2009 Nat.Rev.Immunol, Lindau D et al 2013 Immunology) and its importance in metastatic spread are well established (Steele et al 2016 Cancer Cell, Sano et al 2019 Oncogenesis, Acharyya et al 2012 Cell). Altogether, this information led us to identify the CXCR2 axis as a potential underlying mechanistic target of CRD.

Major

The fact that the CXCR2 inhibition works to the same degree in jet-lagged mice as in non jet-lagged mice suggests that this pathway is not selective to circadian regulation of metastatic behavior in these mice. If it is not, then we are still lacking an explanation for the main observation of the paper. The discussion is not sufficient to explain this fact.

We did not claim that CXCR2 driven metastatic processes are selective or specific to CRD induced alterations. This mechanism is well described and reported previously in normal circadian conditions (Steele et al 2016 Cancer Cell, Sano et al 2019 Oncogenesis, Acharyya et al 2012 Cell). Here we showed that CRD leads to an increased metastatic spread associated with an upregulation of the CXCR2/CXCL5 axis and consequently CXCR2 driven metastatic burden. The use of CXCR2 inhibitor in JL mice helps to decrease cancer cell dissemination and metastasis formation and even though CXCR2 inhibition presents similar effects in LD mice, we propose that CXCR2 inhibition is important to counteract the adverse effects of JL and to slow down tumour progression. The use of CXCR2 inhibition could help when used in combination with conventional chemotherapy to improve therapy outcome in patients with circadian rhythm disrupted tumours.

Major novelties of our study: we showed that CRD increases metastatic spread and we described potential underlying mechanisms: 1) increased stemness and tumour initiation capacity and 2) immunosuppressive shift in the tumour microenvironment driven at least by enhanced CXCR2 axis.

Reviewers' comments:

Reviewer #3 (Remarks to the Author):

The majority of the issues raised by this reviewer were acceptably addressed. This includes comments on Figures 1-4, and Figures 5E-F.

However, some issues remain (Figure 5 – flow data) and Figure 6. In addition, the authors present valuable arguments only in the rebuttal for some points raised by this reviewer, but these arguments have not been incorporated in the main text. The prime example of this is the prominent place the blood cytokine screen still has in the main text (although the data was sent to supplementary). The obvious explanation authors use in their rebuttal, about levels in the blood not been as informative as levels in the primary tumors, has not been used to justify looking at the gene expression levels (but it should be).

Immunophenotyping.

1. The paper used to define the gating strategy for the different immune populations is not an obvious one. Indeed, as the authors recognized, quite unorthodox. This was explained to the reviewer, yet it is missing from the manuscript. Please add this reference in line #211 and make it very clear to the reader. Moreover, remove the references from Movahedi and Franklin (#30 and #31), because leaving them in that section is extremely misleading to the unaware reader. Moreover, ref#30 (Movahedi et al) is used to justify pro- and anti-tumor TAM definition, but it is incorrect, since TAMs are not defined in the same way in this work. Again misleading and inappropriate.

2. Some of the populations do not follow the gating strategy proposed in the Ye et al 2016 paper, critically, the TAM cell population. As per the paper of reference:

I. TAMs should be defined as CD64+CD24+ or CD24- > here they are simply defined as CD24-, therefore including CD64- cells and excluding most of the CD24+ cells. From there on, the remaining derived populations need to be fixed (including MHCII high and low)

II. B cells should be CD24+ > here all "B" cells are CD24 negative

III. Eosinophils are CD11B+ cells > here, the Eos gate includes both CD11B + and CD11B negative cells.

IV. Monocytes should be CD64+ CD11B+ > here, the gating is covered by legends and it's difficult to appreciate, but the CD64 stain is extremely dim and the populations look very different than the same tumors gated in the proposed Ye et al reference.

CXCR2 inhibition.

Without a direct comparison between JL and LD conditions, the claim that inhibition of CXCR2 is responsible for the shift in the immune tumor microenvironment is difficult to make. If the treatment inhibits metastasis in both conditions to a similar degree, which it does, then it is not clear what is specific to CRD conditions. The argument that this treatment could provide "a novel therapeutic tool to thwart the effects of CRD on tumor progression" is flawed if there is no CRD-specificity. In other words, any LD-unrelated treatment that affects metastasis could have been used with the same results. Short of showing a more pronounced effect of the CXCR2 inhibition under JL conditions, I don't see how CXCR2 can be made responsible for CRD-dependent effects.

We sincerely thank the last reviewer for helping us to correct our flow cytometry data and thus to improve the quality and credibility of our research.

Reviewer #3 (Remarks to the Author):

The majority of the issues raised by this reviewer were acceptably addressed. This includes comments on Figures 1-4, and Figures 5E-F.

However, some issues remain (Figure 5 – flow data) and Figure 6. In addition, the authors present valuable arguments only in the rebuttal for some points raised by this reviewer, but these arguments have not been incorporated in the main text. The prime example of this is the prominent place the blood cytokine screen still has in the main text (although the data was sent to supplementary). The obvious explanation authors use in their rebuttal, about levels in the blood not been as informative as levels in the primary tumors, has not been used to justify looking at the gene expression levels (but it should be).

To clarify this point, we included this sentence in the main text:

“Since plasma cytokine/chemokine levels are not necessarily representing the tumour microenvironment, we used the data of our transcriptomic study and real-time PCR to assess the expression levels of cytokines/chemokines and their receptors in primary tumours.”

We decided to mention the Luminex assay in the main text despite the lack of statistical differences between conditions because we think that this data still has information value. And taken that there are no marked differences between LD and JL mice this data was presented in the supplementary information from the beginning.

Immunophenotyping.

1. The paper used to define the gating strategy for the different immune populations is not an obvious one. Indeed, as the authors recognized, quite unorthodox. This was explained to the reviewer, yet it is missing from the manuscript. Please add this reference in line #211 and make it very clear to the reader. Moreover, remove the references from Movahedi and Franklin (#30 and #31), because leaving them in that section is extremely misleading to the unaware reader. Moreover, ref#30 (Movahedi et al) is used to justify pro- and anti-tumor TAM definition, but it is incorrect, since TAMs are not defined in the same way in this work. Again misleading and inappropriate.

The gating strategy is now clearly stated in the main text and referenced throughout.

As suggested we removed the reference #31 Franklin et al. However, we still use the Movahedi et al paper to justify the pro and anti tumour TAM definition. Accordingly, we re-phrased the text and refer to Movahedi et al. where the proper referencing can not be misleading. Also we adjusted our gating strategy: we identified CD11b+MHCII^{hi} and CD11b+MHCII^{low} TAMs on the CD64+CD24⁻ macrophage population. In this way the identified TAMs are phenotypically closer to the MHCII^{low} and MHCII^{hi} macrophages described by Movahedi et al. (where MHCII was the prominent marker to identify TAMs in CD45+CD11b⁺ population)

These new data are shown on the modified Figure 5C and 5D and Supplementary Figure 8 (and see below).

2. Some of the populations do not follow the gating strategy proposed in the Ye et al 2016 paper, critically, the TAM cell population. As per the paper of reference:

I. TAMs should be defined as CD64+CD24+ or CD24- > here they are simply defined as CD24-, therefore including CD64- cells and excluding most of the CD24+ cells. From there on, the remaining derived populations need to be fixed (including MHCII high and low)

We corrected our gating regarding CD64. According to this, we defined macrophages as CD64+CD24- cells. Gating on this population we identified anti- and pro-tumour macrophages as CD11b+MHCIIhi or CD11b+MHCIIlow TAMs respectively. All represented data was corrected (Fig. 5B-D, Supp.Fig 7 and 8).

II. B cells should be CD24+ > here all "B" cells are CD24 negative

Indeed, we did not observe similar CD24 staining on the suspected B cell population as was reported by Yu et al. However, it is important to note that B cells are constitutively expressing MHCII, while murine T cells do not express this marker. Therefore, we believe that MHCII alone is sufficient to differentiate between these two populations.

III. Eosinophils are CD11B+ cells >here, the Eos gate includes both CD11B + and CD11B negative cells.

Gate was corrected and corresponding data was updated (Figure 5B).

IV. Monocytes should be CD64+ CD11B+ >here, the gating is covered by legends and it's difficult to appreciate, but the CD64 stain is extremely dim and the populations look very different than the same tumors gated in the proposed Ye et al reference.

Legends were rearranged, the gating is clearly visible now.

Indeed, in our hand these populations look different but we consistently observed this profile. Please find below a figure showing the mentioned plot compared to the isotype control. Here you can see that we have obvious positive staining (respective gated populations overlaid with isotype control).

Potentially it might be that using different fluorochrome combinations results in different resolutions of the targeted populations. Observing a prominent third population ('CD11c+T cells) might be explained by the significant difference in sampling time and consequent tumour stage: in Yu et al study the analysed tumours were harvested at much later stage (around 6 months/24 weeks of age compared to 16 weeks of age in our study).

CXCR2 inhibition.

Without a direct comparison between JL and LD conditions, the claim that inhibition of CXCR2 is responsible for the shift in the immune tumor microenvironment is difficult to make. If the treatment inhibits metastasis in both conditions to a similar degree, which it does, then it is not clear what is specific to CRD conditions. The argument that this treatment could provide “a novel therapeutic tool to thwart the effects of CRD on tumor progression” is flawed if there is no CRD-specificity. In other words, any LD-unrelated treatment that affects metastasis could have been used with the same results. Short of showing a more pronounced effect of the CXCR2 inhibition under JL conditions, I don't see how CXCR2 can be made responsible for CRD-dependent effects.

We agree that the small number of mice used in our experimental design using the CXCR2 inhibitor does not allow us to observe clear differences between JL and LD mice.

Indeed we can only conclude that the shift in immune microenvironment (improved CD4/CD8 ratio and reduced myeloid infiltration) upon CXCR2 inhibition is present in both conditions, like described previously by other groups (Jamieson T et al. 2012 JCI, Highfill SL et al 2014 Sci Transl Med, Katoh H et al 2013 Cancer Cell, ref#38,39).

But we provided data showing enhanced accumulation of suppressor myeloid and T cells in primary tumours and increased dissemination under JL. Which suggests that the CXCR2 inhibition in combination with conventional therapy could be more beneficial (but not exclusively) to patients with CRD / CRD tumours where the CXCR2 driven mechanisms are accelerated.

We re-write our conclusion about CXCR2 inhibition in CRD tumours and made it clear that it's effect is not CRD specific but it could be especially beneficial for patients with CRD.

REVIEWERS' COMMENTS:

Reviewer #3 (Remarks to the Author):

Thanks to the authors for addressing the remaining points. The data is now more accurate, and the conclusions have been edited appropriately.

We sincerely thank all the reviewers for helping us to improve the quality of this manuscript.

REVIEWERS' COMMENTS:

Reviewer #3 (Remarks to the Author):

Thanks to the authors for addressing the remaining points. The data is now more accurate, and the conclusions have been edited appropriately.

No additional issues were raised by any of the reviewers.

We just want to mention, that we have to correct the statistical tests for Fig.2c-d, Fig.4f, Fig. 6e, supplementary Fig10b. Indeed, by compiling the Source data file, we saw that we used percentage instead of numbers for calculating p-values. After correction using a binomial test, the p-values are higher but still below the threshold of 0.05 (except for supplementary Fig.10b).